# Long non-coding RNA-dependent mechanism to regulate heme biosynthesis and erythrocyte development

Jinhua Liu[1,2], Yapu Li[1,2], Jingyuan Tong[1,2], Jie Gao[1,2], Qing Guo[1,2], Lingling Zhang[3], Bingrui Wang[1,2], Hui Zhao[3], Hongtao Wang[1,2], Erlie Jiang[1,2], Ryo Kurita[4], Yukio Nakamura[5], Osamu Tanabe[6], James Douglas Engel[7], Emery H. Bresnick[8], Jiaxi Zhou[1,2] & Lihong Shi[1,2]

In addition to serving as a prosthetic group for enzymes and a hemoglobin structural component, heme is a crucial homeostatic regulator of erythroid cell development and function. While lncRNAs modulate diverse physiological and pathological cellular processes, their involvement in heme-dependent mechanisms is largely unexplored. In this study, we elucidated a lncRNA (UCA1)-mediated mechanism that regulates heme metabolism in human erythroid cells. We discovered that UCA1 expression is dynamically regulated during human erythroid maturation, with a maximal expression in proerythroblasts. UCA1 depletion predominantly impairs heme biosynthesis and arrests erythroid differentiation at the proerythroblast stage. Mechanistic analysis revealed that UCA1 physically interacts with the RNA-binding protein PTBP1, and UCA1 functions as an RNA scaffold to recruit PTBP1 to *ALAS2* mRNA, which stabilizes *ALAS2* mRNA. These results define a lncRNA-mediated post-transcriptional mechanism that provides a new dimension into how the fundamental heme biosynthetic process is regulated as a determinant of erythrocyte development.

[1] State Key Laboratory of Experimental Hematology, Institute of Hematology and Blood Diseases Hospital, Chinese Academy of Medical Sciences & Peking Union Medical College, Tianjin 300020, China. [2] Center for Stem Cell Medicine, Chinese Academy of Medical Sciences, Beijing 100730, China. [3] Tianjin Key Laboratory of Food and Biotechnology, School of Biotechnology and Food Science, Tianjin University of Commerce, Tianjin 300134, China. [4] Japanese Red Cross Society, Department of Research and Development, Central Blood Institute, Tokyo 105-8521, Japan. [5] RIKEN BioResource Research Center, Cell Engineering Division, Ibaraki 305-0074, Japan. [6] Department of Integrative Genomics Tohoku Medical Megabank, Tohoku University, Sedai 980-8573, Japan. [7] Department of Cell and Developmental Biology, University of Michigan Medical School, Ann Arbor, MI 48109, USA. [8] Wisconsin Institutes for Medical Research, Paul Carbone Cancer Center, Department of Cell and Regenerative Biology, University of Wisconsin School of Medicine and Public Health, Madison, WI 53562, USA. These authors contributed equally: Jinhua Liu, Yapu Li, Jingyuan Tong. Correspondence and requests for materials should be addressed to J.Z. (email: zhoujx@ihcams.ac.cn) or to L.S. (email: shilihongxys@ihcams.ac.cn)

As a prosthetic group, heme is involved in diverse biological processes, such as electron transfer and oxygen transport[1,2]. In addition, heme is a vital structural component of hemoglobin. Beyond these functions, heme plays crucial regulatory roles during erythroid differentiation by regulating its own synthesis[3,4] and by aiding the erythroid master regulator GATA1 to establish and maintain the erythroblast transcriptome[5]. Given these essential functions, heme biosynthesis defects in erythroblasts can cause pathologies such as sideroblastic anemia or erythropoietic porphyria[4].

Through a series of enzymatic reactions, heme is synthesized in erythroid progenitors beginning with proerythroblasts[6]. The first and rate-limiting step during heme biosynthesis is catalyzed by 5-aminolevulinic acid synthase 2 (ALAS2, also known as ALASE), which is an erythroid-specific gene[7,8]. ALAS2 expression is strongly induced during erythroid differentiation to meet the intense demand for hemoglobin assembly[8,9]. Alas2$^{-/-}$ mutation results in murine embryonic lethality, as it causes severe anemia resulting from arrested erythroid differentiation at the proerythroblast stage due to heme insufficiency[10].

ALAS2 expression is tightly controlled at both the transcriptional and translational levels. Transcriptionally, GATA1 binds canonical GATA motifs within the ALAS2 promoter and introns to activate transcription[5,11–13]. The intron 1 GATA motif anchors the intron 8 GATA motif to the ALAS2 proximal promoter region, thereby forming a long-range enhancer loop to confer high-level ALAS2 transcription[13]. Iron-dependent translational control modulates ALAS2 protein synthesis via the iron-responsive element (IRE)/iron-regulatory protein (IRP) system[14,15]. Under conditions of iron insufficiency, IRPs bind to IRE in the ALAS2 mRNA 5′ untranslated region to inhibit its translation. Although prior work revealed transcriptional and translational mechanisms controlling ALAS2 expression, post-transcriptional mechanisms have not been described.

Long non-coding RNAs (lncRNAs) are critical regulators of protein-coding and non-coding genes and are implicated in diverse physiological and pathological cellular processes[16], including normal and malignant hematopoiesis[17,18]. LncRNAs interact with RNA, DNA, and/or proteins to regulate chromatin modifications, transcription and pre-mRNA splicing and to function as scaffolds for protein complex assembly[19]. Although thousands of erythroid stage-specific lncRNAs have been identified[20–24], only a few have been functionally analyzed. For example, long intergenic non-coding RNA (lincRNA) EPS[25] and lncRNA Fas-antisense 1[26] promote erythroid progenitor survival; lnc-EC1 and lnc-EC6 regulate erythroblast enucleation;[22,27] and lncRNA-αGT confers maximal activation of α-globin gene expression in chickens[28]. Nevertheless, the biological functions of the vast majority of lncRNAs have not been described.

Urothelial carcinoma-associated 1 (UCA1) is a lncRNA that was originally cloned and identified in the bladder cancer cell line BLZ-211[29]. The oncogenic functions of UCA1 have been studied extensively in various cancer types, including bladder cancer, acute myeloid leukemia, breast cancer, colorectal cancer, etc[30]. UCA1 ectopic expression promotes cell proliferation as well as tumor progression, migration and drug resistance, which are mediated by distinct mechanisms[31]. For example, UCA1 increases cell proliferation by regulation of ELF4 and its downstream targets KRT6/13 or via FGFR1/ERK signaling pathway in prostate cancer[32] and hepatocellular carcinoma[33], respectively. UCA1 promotes tumor progression by targeting miR-193a-3p and miR-204-5p in non-small cell lung cancer[34] and colorectal cancer[35], respectively. In contrast to its broad expression in cancers, UCA1 expression is restricted to normal heart and spleen tissues, as determined by analyses of 15 tissues in adult humans, including liver, kidney, lung, and others[36]. Although UCA1 oncogenic

functions have received considerable attention, physiological UCA1 functions during development and differentiation are only poorly defined.

Herein, we elucidated a post-transcriptional mechanism involving UCA1-regulated ALAS2 mRNA stability. We demonstrated that UCA1 serves as an essential RNA scaffold to recruit an RNA-binding protein (PTBP1) to ALAS2 mRNA, which confers ALAS2 mRNA stability. When UCA1 or PTBP1 are depleted by lentiviral-mediated shRNAs, ALAS2 mRNA stability declines, and ALAS2 expression is attenuated, thus impairing heme biosynthesis and inhibiting erythroid differentiation. In aggregate, these findings illustrate a new regulatory circuit that mediates heme biosynthesis and erythroid maturation.

## Results

**Differential UCA1 expression during human erythropoiesis.** Human cord blood CD34$^+$ progenitor cells were purified and subjected to erythroid differentiation ex vivo[37]. After 8 days in culture, ~90% of the cells resembled proerythroblasts, and after day 14, the cells began to undergo enucleation, indicative of terminal erythroid maturation (Fig. 1a). Erythroid differentiation was also confirmed by benzidine staining and fluorescence-activated cell sorting (FACS) analysis (Fig. 1b–e).

To discover new regulators of erythrocyte development and function, we conducted RNA-seq analysis with cells on days 4, 8, 11, and 14 during erythroid differentiation. Since non-coding RNAs (ncRNAs), especially lncRNAs, can be important regulators of diverse developmental and differentiation processes, we identified differentially expressed ncRNAs in this system (Supplementary Fig. 1A, left panel). We were particularly interested in ncRNAs that were upregulated at the proerythroblast stage, when robust globin and heme biosynthesis begin (Supplementary Fig. 1A, B). Among these, lncRNA UCA1 was one of the most abundantly and dynamically expressed genes (Supplementary Fig. 1A, right panel and Supplementary Fig. 1B). Quantitative real-time PCR (qRT-PCR) analysis confirmed that UCA1 expression was maximal on day 8 of differentiation when the majority of cells are proerythroblasts (Fig. 1f). Our results were consistent with a prior analysis of the transcriptome of FACS-sorted, stage-specific erythroid cells differentiated from human cord blood CD34$^+$ progenitor cells (Supplementary Fig. 1C)[23]. A similar UCA1 expression pattern was also detected in erythroid cells differentiated from adult peripheral blood CD34$^+$ cells (Supplementary Fig. 1D)[24].

Intriguingly, an evolutionary analysis of UCA1 nucleotide sequence using the phastcons and phyloP algorithms and sequences from the UCSC database did not reveal conservation between human and mouse (Fig. 1g). UCA1 showed low conservation in monkeys, but high conservation in great apes, such as gorillas and chimpanzees (which are phylogenetically most closely related to humans), suggesting that it might be a primate-specific lncRNA.

**UCA1 depletion blocks erythroid maturation by impairing heme metabolism.** To assess the physiological function of UCA1, we infected lentiviral-mediated UCA1-targeting short hairpin RNA (shRNA) into cells differentiated for 4 days from cord blood CD34$^+$ cells. UCA1 expression was significantly reduced on days 8 and 14 of differentiation after infection of the shRNAs (Fig. 2a). FACS analysis revealed that the percentage of CD71$^+$CD235$^+$ cells decreased significantly, suggesting that UCA1 reduction impaired erythroid maturation (Fig. 2b, c). Wright-Giemsa staining provided evidence for accumulation of cells with the hallmark attributes of proerythroblasts, including large cell size, high nucleus/cytoplasm ratio, prominent nucleoli and intensely

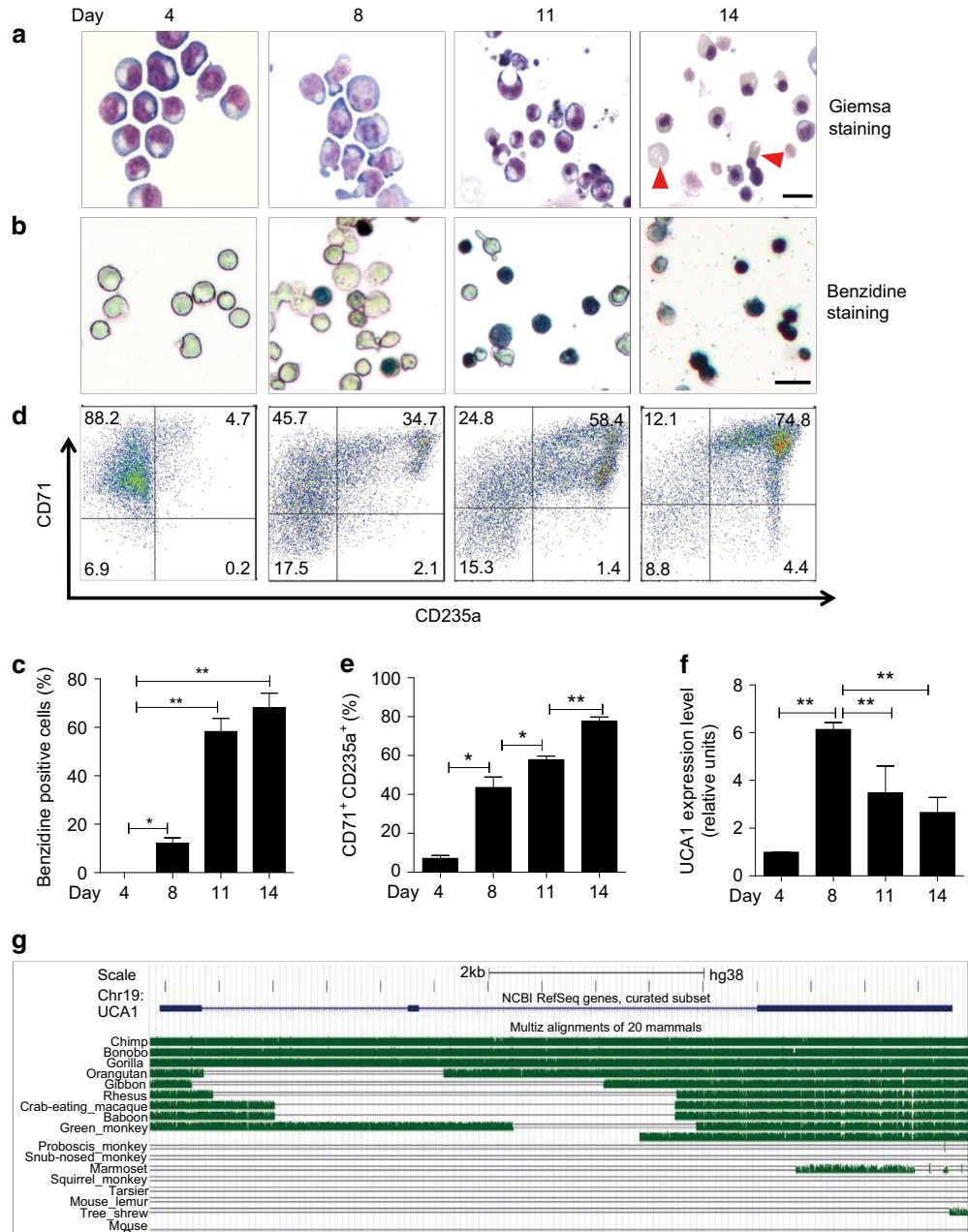

**Fig. 1** UCA1 expression peaks in proerythroblasts during human erythropoiesis. **a** Primary human erythroid cells differentiated from hematopoietic cord blood progenitor CD34[+] cells ex vivo were monitored by Wright-Giemsa staining on days 4, 8, 11, 14. Arrowheads denote enucleated reticulocytes. Scale bar = 10 μm. **b**, **c** Benzidine staining (**b**) shows the hemoglobin production at the indicated times. Scale bar = 20 μm. The bar graph (**c**) depicts the percentage of benzidine-positive cells. **d**, **e** Flow-cytometric analysis (**d**) of the expression of differentiation markers CD71 and CD235a at the indicated times. The bar graph (**e**) depicts the percentage of CD71[+]CD235a[+] cells. **f** The relative UCA1 abundance was quantitated by qRT-PCR during primary erythroid differentiation. The level of UCA1 RNA was normalized to that of *18 S* rRNA. **g** *UCA1* conservation among mammals from UCSC Genome Browser. Bar graphs were generated with data from three independent experiments (*n* = 3). Error bars represent SEM. *P*-values were determined by Student's *t*-test. *$P < 0.05$, **$P < 0.01$

basophilic cytoplasm[38] (Fig. 2d). Evidence for a differentiation blockade resulting from the UCA1 knockdown included accumulation of larger cells and nuclei (Fig. 2e, f), reduced hemoglobin production (Fig. 2g, h) and reduced enucleation (Supplementary Fig. 2A, B).

To identify downstream targets of UCA1, we conducted RNA-seq analysis in cultured erythroid cells on day 8 of differentiation with or without UCA1 downregulation (Fig. 2i). This analysis identified 2542 differentially expressed genes (DEGs) (cutoffs: $P_{adj} < 0.05$ and FPKM > 0.1). We then applied hallmark enrichment analysis, which summarizes and represents specific well-

defined biological states or processes, and found the DEGs were enriched for multiple cellular processes and signaling pathways, including heme metabolism, E2F targets, mitotic spindle, and others (Fig. 2j). Furthermore, as is consistent with the hallmark enrichment analysis, gene set enrichment analysis (GSEA)[39] showed that the heme metabolic pathway was significantly impaired after downregulation of UCA1 as well (Supplementary Fig. 2C). Despite the erythroid maturation blockade, key erythroid transcriptional regulators (KLF1, LMO2, GATA1, and TAL1), cytoskeletal components (EPB41) and hemoglobin subunits (α- and γ-globin) exhibited little to no change after

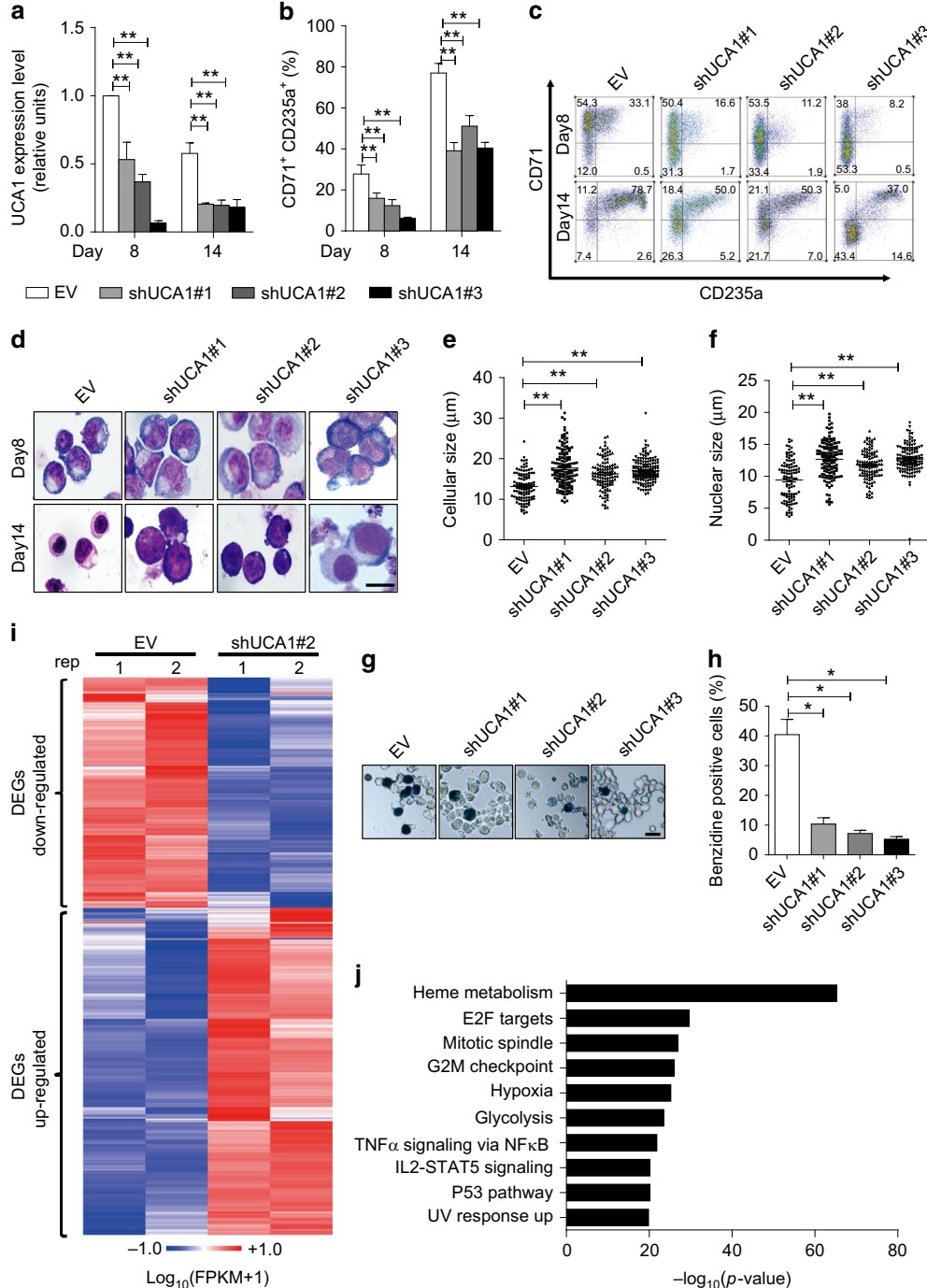

**Fig. 2** UCA1 depletion impairs heme metabolism and blocks erythroid maturation. **a** The relative UCA1 expression was quantitated by qRT-PCR on days 8 and 14 in shRNA or control lentivirus (EV) infected primary erythroid cells. *18S* rRNA was used as the internal control. **b–h** Human primary erythroid differentiation was monitored by flow cytometry (CD71+CD235a+ cells) (**b**, **c**), Wright-Giemsa staining (**d**, Scale bar = 10 μm), the cellular (**e**) and nuclear size (**f**) (on day 14, *n* = 100), and benzidine staining (**g**, **h**, Scale bar = 20 μm) on days 14 in shRNA lentivirus infected cells or control lentivirus (EV) infected cells. **i** The heatmap depicts DEGs profiling after UCA1 depletion (shRNA#2) in differentiated erythroblasts at day 8 ($P_{adj} < 0.05$ and FPKM > 0.1). **j** Hallmark gene set enrichment analysis of DEGs after UCA1 knockdown. Bar graphs were generated with data from three independent experiments (*n* = 3). Error bars represent SEM. *P*-values were determined by Student's *t*-test.*$P < 0.05$, **$P < 0.01$

UCA1 depletion, except for adult β-globin (Supplementary Fig. 2D, E).

In aggregate, these results provide evidence that UCA1 controls erythroid maturation. When UCA1 becomes limiting, erythroid precursors are unable to progress beyond the proerythroblast stage. Since heme biosynthesis and the control of heme levels are important determinants of erythrocyte development and

function, it is instructive to investigate potential links between UCA1 and heme-dependent mechanisms.

**UCA1 interacts with the RNA-binding protein PTBP1.** Since the subcellular localization of a lncRNA can provide functional insights into its function[40], we utilized a highly sensitive and

specific RNA in situ hybridization method (RNAscope® multiplex fluorescence)[41] to detect the UCA1 cellular distribution in primary erythroid cells on day 8 of differentiation. This analysis revealed that UCA1 is expressed and distributed predominantly in the cytoplasm of the erythroid cells (Fig. 3a and Supplementary Fig. 3A, B). Here, peptidylprolyl isomerase B (PPIB) and the bacterial gene dapB were used as positive and negative controls,

respectively. The specificity of UCA1 probes was further validated in mouse erythroleukemia cells (MELs), where no signal was detected (Fig. 3a). As cytoplasmic lncRNAs usually interact with RNA-binding proteins to execute their cellular functions[42], we conducted an RNA pull-down assay using an in vitro transcribed UCA1 and cytoplasmic lysates from AraC-induced K562 cells, combined with mass spectrometry (MS) analysis, to identify

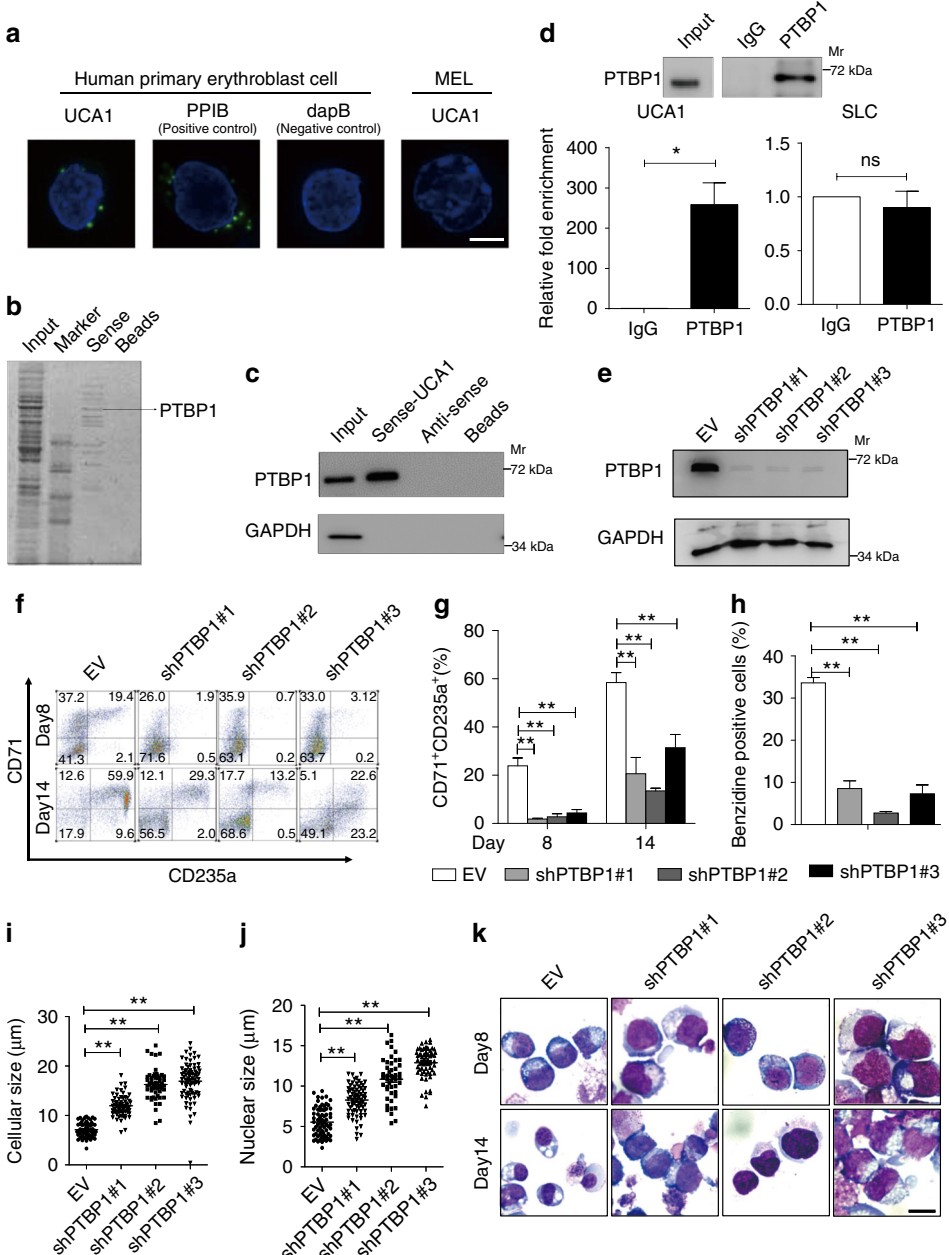

**Fig. 3** UCA1 interacts with PTBP1 to regulate erythroid maturation. **a** UCA1 localization was determined by RNA-FISH in primary erythroid cells differentiated ex vivo for 8 days. Probes for *PPIB* act as a positive control and probes for bacterial gene *dapB* act as a negative control. MEL cells act as another negative control for the specificity of *UCA1* probes. Scale bar = 5 μm. **b** Mass spectrometry identified UCA1-interacting proteins after UCA1 RNA pull-down assay using in vitro transcribed RNA in AraC-induced K562 cells. **c** WB using anti-PTBP1 antibody, followed by UCA1 and antisense-UCA1 pull-down in HUDEP-2 cells. GAPDH was used as a negative control. **d** PTBP1 RIP assay in HUDEP-2 cells. WB confirmed PTBP1 immunoprecipitation (top). The relative fold enrichment of UCA1 in comparison with IgG was determined by qRT-PCR (bottom). SLC25A21-AS1 (hereafter we called it SLC for short) is an unrelated lncRNA and acts as a negative control. **e** PTBP1 expression examined by WB after shRNA-mediated PTBP1 knockdown with day 8 differentiated primary erythroid cells. GAPDH was used for internal control. **f–k** Human primary erythroid differentiation after PTBP1 depletion was assessed by flow cytometry (CD71⁺CD235a⁺ cells) (**f**, **g**), benzidine staining (percentage of positive cells) (**h**), the cellular and nuclear size (**i**, **j**) (on day 14, n = 50) and Wright-Giemsa staining (**k**, Scale bar = 10 μm) after 8 and 14 days induction. Bar graphs were generated with data from three independent experiments (n = 3). Error bars represent SEM. P-values were determined by Student's t-test. *P < 0.05, **P < 0.01

potential UCA1-associated proteins. This analysis identified PTBP1 as a possible UCA1-interacting protein (Fig. 3b). This interaction was validated by in vitro UCA1 RNA pull-down assay, followed by immunoblotting with anti-PTBP1 antibody in AraC-induced K562 cells (Supplementary Fig. 3C) and in an immortalized human umbilical cord blood CD34$^+$-derived erythroid progenitor cell line (HUDEP-2)[43] (Fig. 3c), which resembles proerythroblasts. Simultaneously, RNA immunoprecipitation (RIP) using anti-PTBP1 antibody, followed by qRT-PCR, detected UCA1 enrichment in AraC-induced K562 (Supplementary Fig. 3D) and HUDEP-2 cells (Fig. 3d).

PTBP1, also termed HNRNP1 or PTB, belongs to a subfamily of heterogeneous nuclear ribonucleoproteins, which contains RNA-binding motifs involved in mRNA metabolic processes including mRNA stabilization, alternative splicing, internal ribosome entry site (IRES)-mediated translation initiation and 3′-end processing among others[44]. To elucidate PTBP1 function during erythroid differentiation, we lentivirally infected PTBP1-targeted shRNAs into erythroid cells differentiated for 4 days from cord blood CD34$^+$ cells. The decreased expression of PTBP1 was confirmed at both RNA (Supplementary Fig. 3E) and protein (Fig. 3e) levels. Downregulating PTBP1 reduced the number of CD71$^+$CD235a$^+$ erythroblasts (Fig. 3f, g), decreased globin synthesis (Fig. 3h), and increased cellular and nuclear size (Fig. 3i, j). Giemsa staining revealed an accumulation of cells at the immature erythroblast stage (Fig. 3k). In summary, UCA1 physically interacts with PTBP1, and their similar loss-of-function phenotypes support the notion that UCA1 functions with PTBP1 to control human erythroid maturation.

**Linking UCA1 and PTBP1 to heme biosynthesis**. To identify downstream targets of PTBP1 and UCA1, we conducted RNA-seq analysis using erythroid cells recovered from day 8 CD34$^+$ erythroid differentiation cultures, with or without PTBP1 downregulation (cutoffs: $P_{adj} < 0.05$ and FPKM > 0.1, Supplementary Fig. 4A). Hallmark enrichment analysis revealed that the DEGs after PTBP1 downregulation were associated with heme metabolism, E2F targets, and more (Supplementary Fig. 4B). GSEA also showed that genes correlated with heme metabolism were significantly downregulated upon PTBP1 reduction (Supplementary Fig. 4C).

We identified the common DEGs (1355 genes) altered by both UCA1 and PTBP1 (Fig. 4a and Supplementary Fig. 4D). Hallmark enrichment analysis and GSEA revealed that these common DEGs were most significantly enriched in heme metabolism (Fig. 4b, c). This correlation analysis suggests that the physical interaction between UCA1 and PTBP1 modulates erythroid maturation, and it is attractive to consider the potential functional importance of heme biosynthesis to the UCA1/PTBP1 mechanism (Fig. 4b–d).

To further clarity which pathway within heme metabolism is influenced by UCA1 and PTBP1, the common DEGs related to heme metabolism were extracted and subjected to Gene Ontology (GO) enrichment analysis (Fig. 4d, e). This analysis revealed that heme biosynthesis was the most significantly affected pathway (Fig. 4e). The downregulated genes related to heme biosynthesis after UCA1 or PTBP1 knockdown were further validated by qRT-PCR (Fig. 4f, g).

**PTBP1 protein, UCA1 RNA, and *ALAS2* mRNA form a protein–RNA complex**. To analyze how the physical interaction between PTBP1 and UCA1 might regulate heme biosynthesis, we tested whether genes involved in heme biosynthesis physically interact with UCA1 and PTBP1 in HUDEP-2 cells. Using a RIP assay with an anti-PTBP1 antibody, we detected PTBP1

interaction with multiple mRNAs expressed from heme biosynthesis-related genes, such as *ALAS2, PPOX*, and *FECH* (Fig. 5a). However, when performing the endogenous RNA–RNA pull-down assay using biotinylated UCA1 DNA probes to detect the UCA1-interacting RNAs, we found that the enrichment of *ALAS2* mRNA was much higher (7.4-Fold higher, $P = 0.01$ determined by Student's $t$-test) than that of other heme biosynthetic RNAs in HUDEP-2 cells (Fig. 5b, c). A reverse RNA–RNA pull-down assay using biotinylated *ALAS2* DNA probes also verified the UCA1-*ALAS2* mRNA interaction (Fig. 5d–f). The interaction between *ALAS2* mRNA and PTBP1 was additionally confirmed by RNA pull-down using in vitro transcribed *ALAS2* mRNA, followed by PTBP1 immunoblotting in HUDEP-2 cell extracts (Fig. 5g). The interaction between UCA1, PTBP1 and *ALAS2* mRNA was validated in AraC-induced K562 cells as well (Supplementary Fig. 5). These results provide direct evidence that PTBP1 protein, UCA1 RNA, and *ALAS2* mRNA physically interact and suggest that these interactions reflect a protein–RNA complex involving all three components.

To identify *ALAS2* mRNA sequences required for PTBP1 binding, we in vitro transcribed three *ALAS2* mRNA fragments, including full-length *ALAS2* mRNA, the 5′ two-thirds (F1) and the 5′ half (F2) (Fig. 5h). We conducted RNA pull-down analyses with these *ALAS2* mRNA fragments, followed by PTBP1 immunoblotting in AraC-induced K562 cells. Although the full-length and F1 fragments bound PTBP1, binding was not detected with the F2 fragment (Fig. 5i), suggesting that the 3′ fragment of *ALAS2* mRNA may be important for PTBP1 binding. We further dissected the 3′end and found that *ALAS2* mRNA (5′ 850–1100 nt, indicated by the dashed lines), which contains the PTBP1 canonical binding motif CUCC[45], associated with PTBP1 (Fig. 5h, j). To test whether the binding site is a critical mediator of the interaction between PTBP1 and *ALAS2* mRNA in HUDEP-2 cells, we in vitro transcribed the F4 fragment with the presumptive binding site deleted (F4Δ(CUCC)) (Fig. 5h) and performed the RNA pull-down assay by incubating F4 and F4Δ(CUCC) fragments with HUDEP-2 cytoplasmic extracts. We did not detect association with the F4Δ(CUCC) fragment although an intense signal was recovered when examining the F4 fragment (Fig. 5k), indicating that the CUCC motif is essential for *ALAS2* mRNA and PTBP1 protein interaction. We, therefore, infer that a complex involving PTBP1 protein, UCA1 and *ALAS2* mRNA confers ALAS2 gene regulation, as *ALAS2* mRNA (Fig. 4f, g) and protein (Fig. 5l, m) were significantly reduced in day 8 differentiated primary erythroid cells after UCA1 and PTBP1 downregulation.

**UCA1 is indispensable for PTBP1-mediated *ALAS2* mRNA stability**. Since the function of PTBP1 in the cytoplasm is known to regulate mRNA stability[44], we investigated whether PTBP1 and UCA1 regulate ALAS2 mRNA levels by influencing its stability. Actinomycin D (ACD) was used to block de novo transcription in AraC-induced K562 cells, which were infected with shRNA of UCA1 and PTBP1, alone or in combination. *ALAS2* mRNA loss increased under conditions in which UCA1 or PTBP1 were downregulated (Fig. 6a–c). The concomitant downregulation of UCA1 and PTBP1 also decreased *ALAS2* mRNA stability (Fig. 6c). These results indicate that a ternary protein–RNA complex post-transcriptionally stabilizes *ALAS2* mRNA.

We next tested whether ALAS2 overexpression could rescue the erythroid differentiation delay caused by UCA1 depletion. We utilized CRISPR/Cas9 to generate K562 UCA1 knockout cell lines ($UCA1^{-/-}$) with two sgRNAs targeted to the 5′ and 3′ ends of *UCA1*, respectively (Fig. 6d, top panel). However, we were unable to recover $UCA1^{-/-}$ cell lines from multiple clones. Instead, we

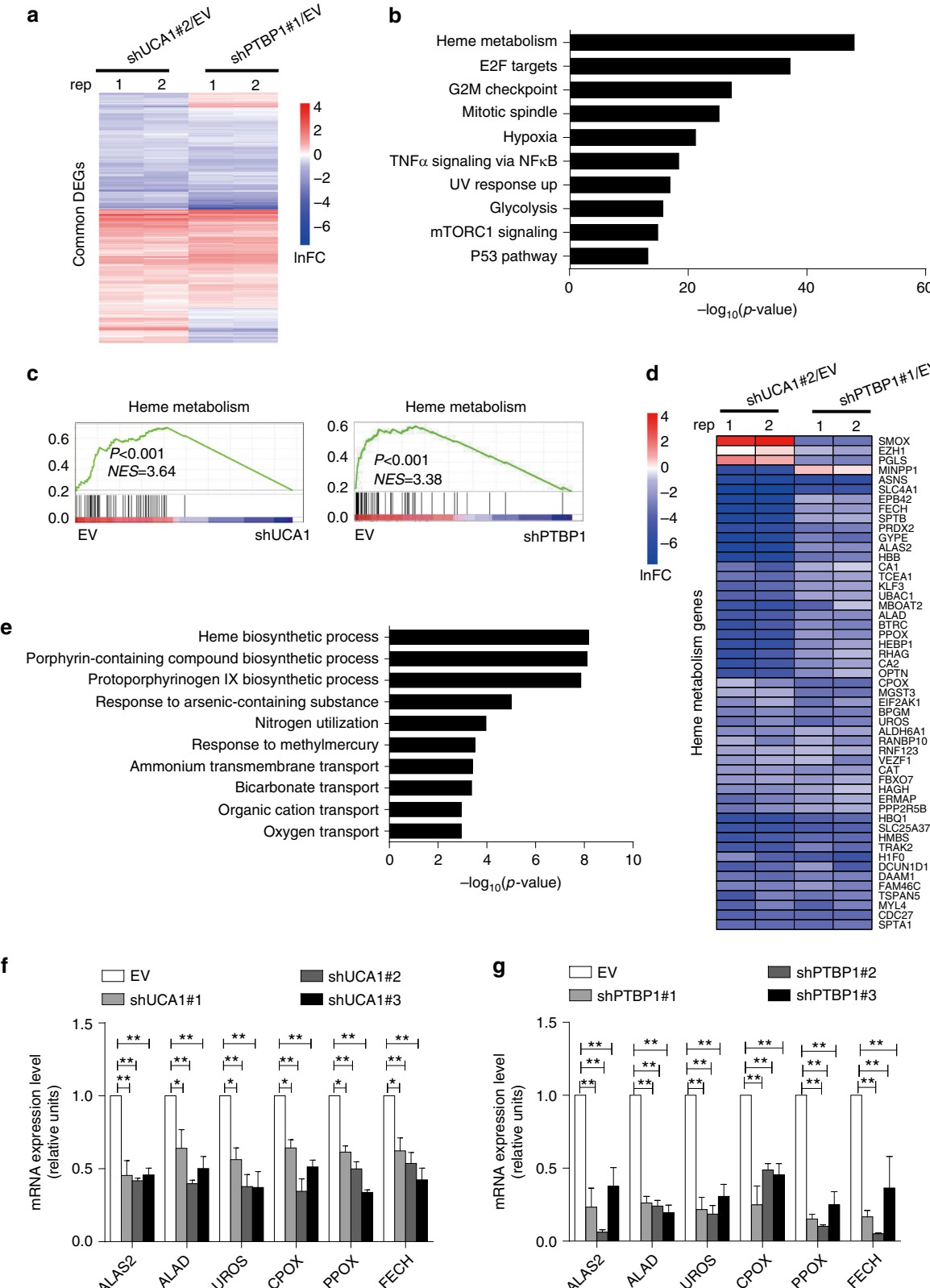

**Fig. 4** Linking UCA1 and PTBP1-regulated erythroid maturation to the control of heme biosynthesis. **a** The heatmap depicts the fold-change in DEGs after UCA1 (shRNA#2/EV) or PTBP1 depletion (shRNA#1/EV) in differentiated erythroblasts at day 8. **b** Hallmark enrichment analysis of the common DEGs. **c** GSEA showed heme metabolism gene set enrichment from the common DEGs after UCA1 (left) or PTBP1 (right) depletion. Normalized enrichment scores (NES) and *P*-values are indicated in each plot. **d** Heatmap shows the expression of heme metabolism related genes after UCA1 or PTBP1 downregulation. **e** GO analysis of the common heme metabolism related genes after UCA1 or PTBP1 depletion. **f, g** qRT-PCR to analyze the expression of genes involved in heme biosynthesis on days 8 in differentiated primary erythroid cells after UCA1 (**f**) or PTBP1 (**g**) depletion. *GAPDH* mRNA was used as an internal control. Bar graphs were generated with data from three independent experiments (*n* = 3). Error bars represent SEM. *P*-values were determined by Student's *t*-test. *$P < 0.05$, **$P < 0.01$

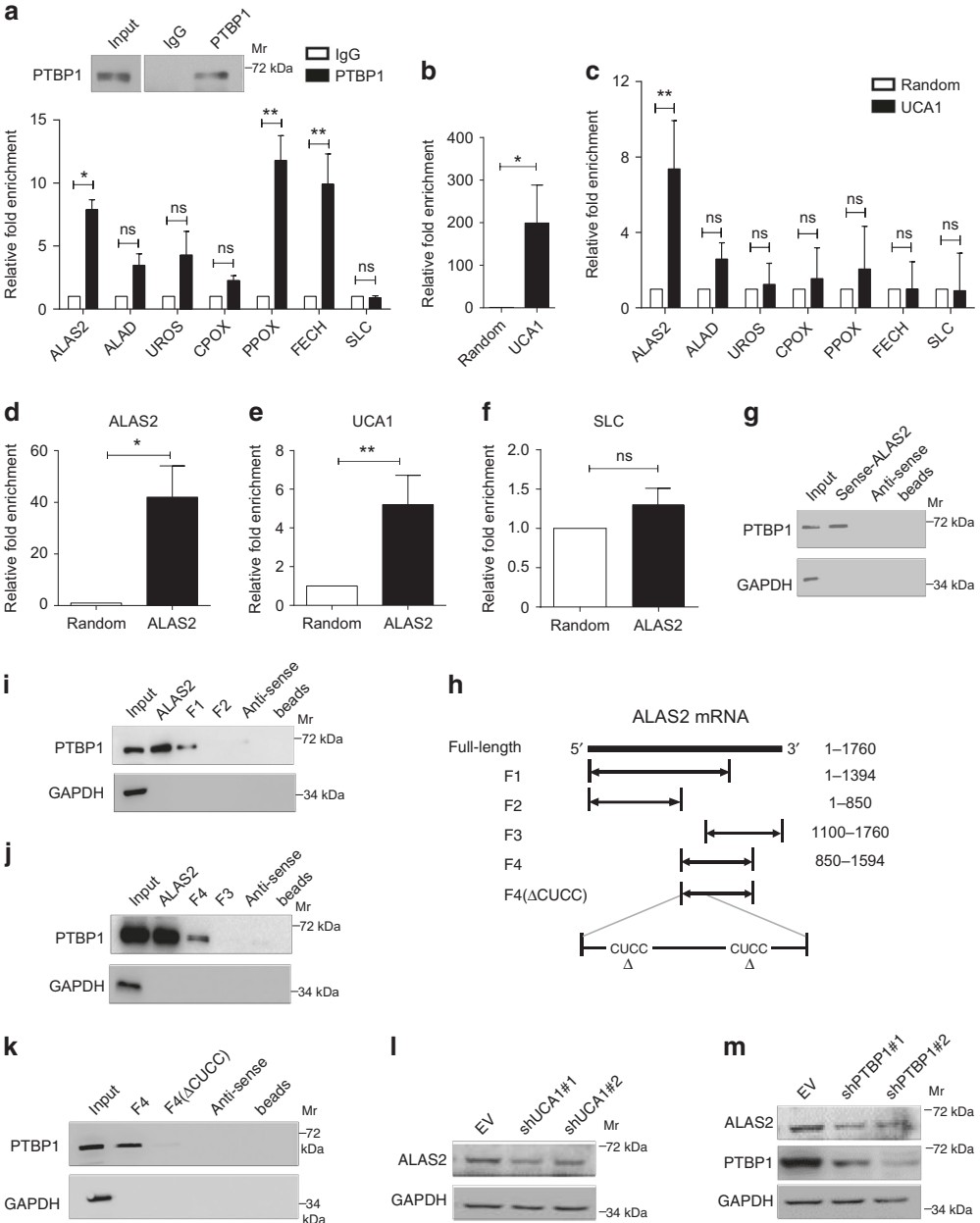

**Fig. 5** PTBP1 protein, UCA1 RNA, and ALAS2 mRNA form a protein–RNA complex. **a** PTBP1 RIP assay to analyze interactions between PTBP1 and heme biosynthesis-related genes in HUDEP-2 cells. WB shows PTBP1 immunoprecipitation (top). The relative fold enrichment of heme biosynthesis-related mRNAs compared to IgG was determined by qRT-PCR (bottom). SLC was used as a negative control. **b, c** The enrichment of UCA1 (**b**) and heme biosynthetic mRNAs detected by in vivo RNA–RNA pull-down assay using *UCA1*-specific probes and a random probe in HUDEP-2 cells. SLC was used as a negative control. **d–e** The enrichment of *ALAS2* mRNA (**d**), UCA1 (**e**) and SLC (**f**) by the in vivo RNA–RNA pull-down assay using *ALAS2*-specific probes in HUDEP-2 cells. SLC was used as a negative control. **g** WB using anti-PTBP1 antibody followed by in vitro transcribed *ALAS2* and antisense-*ALAS2* mRNA pull-down assay in HUDEP-2 cells. GAPDH was used as a negative control. **h–k** In vitro transcribed full-length or fragmental *ALAS2* mRNA pull-down assays followed by PTBP1 immunoblotting in AraC-induced K562 cells (**i**, **j**) and HUDEP-2 cells (**k**). We have in vitro transcribed full-length *ALAS2*, from 5′ F1 (1–1394 nt), F2 (1–850 nt), F3 (1100–1760 nt), F4 (850–1594 nt), F4 fragment with PTBP1-binding motif deletion (F4(ΔCUCC)) and full-length antisense-*ALAS2* mRNA. GAPDH was used as a negative control. **l, m** WB shows ALAS2 expression after UCA1 (**l**) and PTBP1 (**m**) depletion in primary erythroid cells differentiated for 8 days. GAPDH was used as a loading control. $n = 3$ independent experiments. Error bars represent SEM. *P*-values were determined by Student's *t*-test. *$P < 0.05$, **$P < 0.01$

found one compound heterozygous mutant cell line bearing two mutant alleles, in which one allele was deleted and an indel ($+5/−1$ bp) occurred in the other allele ($UCA1^{+/−}$) (Fig. 6d, bottom panel); in this line, UCA1 expression was significantly reduced (Fig. 6e). *ALAS2* mRNA expression was decreased in the $UCA1^{+/−}$ cells as well (Fig. 6f).

We transfected an ALAS2 expression vector into $UCA1^{+/−}$ cells and induced erythroid differentiation by AraC treatment. The expression of *ALAS2* mRNA (Fig. 6g) and protein (Fig. 6h) was measured by qRT-PCR and western blotting (WB). Over-expression of ALAS2 partially rescued erythroid differentiation in $UCA1^{+/−}$ cells as evidenced by a more intensely red pellet

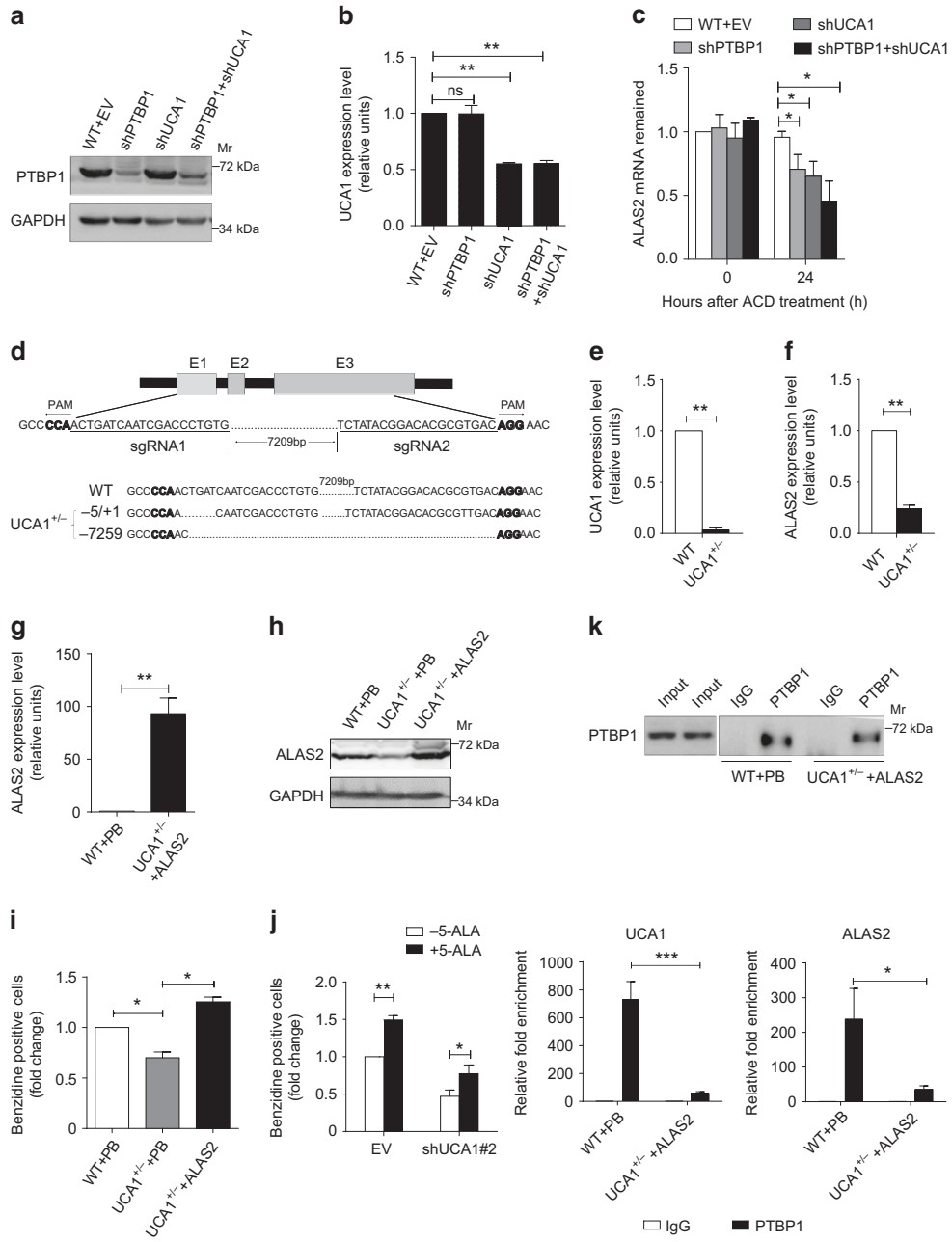

**Fig. 6** UCA1 is indispensable for PTBP1 regulation of *ALAS2* mRNA stability. **a–c** *ALAS2* mRNA stability assessment. K562 cells were infected with lentiviral control vector (EV), sh*UCA1*, shPTBP1, and sh*UCA1* + shPTBP1 after induced by AraC for 24 h. The cells were treated with Actinomycin D (5 µg per ml) for 24 h to quantitate *ALAS2* mRNA stability. The expression of PTBP1 and UCA1 was quantitated by WB (**a**) and qRT-PCR (**b**), respectively. The *ALAS2* mRNA abundance, relative to *GAPDH*, was quantified by qRT-PCR (**c**). **d** Schematic illustrating *UCA1* deletion using CRISPR/Cas9 technology in K562 cells (top). *UCA1* compound heterozygous mutant cell line (*UCA1*$^{+/-}$) was confirmed by DNA sequencing (bottom). **e, f** The relative abundance of UCA1 (**e**) and *ALAS2* (**f**) mRNA was quantified by qRT-PCR in WT and *UCA1*$^{+/-}$ cells, respectively. **g, h** The overexpression of *ALAS2* mRNA (**g**) and protein (**h**) was measure by qRT-PCR and WB, respectively. ALAS2 piggybac transposon expression vector or the empty piggybac vector (PB) were transfected to WT or *UCA1*$^{+/-}$ K562 cells. *GAPDH* mRNA and protein was used for internal control for qRT-PCR and WB, respectively. **i** The relative fold-change of benzidine-positive cells (%) in *UCA1*$^{+/-}$ cells with ALAS2 overexpression (*UCA1*$^{+/-}$ + ALAS2) and *UCA1*$^{+/-}$ cells with empty piggybac expression vector (*UCA1*$^{+/-}$ + PB) to that of WT K562 cells transfected with empty vector (WT + PB). **j** The relative fold-change of benzidine-positive cells (%) at day 14 differentiated primary erythroid cells. The cells were first infected with lentiviral control vector (EV) or UCA1 shRNA#2 at day 4 and then 300 µM 5-ALA was supplemented to the culture medium at day 6. **k** PTBP1 RIP assay was conducted to test for interaction between PTBP1 and *ALAS2* mRNA. WB analysis of PTBP1 immunoprecipitation in WT or *UCA1*$^{+/-}$ cells with ALAS2 overexpression (top). The enrichment of UCA1 RNA and *ALAS2* mRNA in PTBP1 immunoprecipitates was detected by qRT-PCR in WT and *UCA1*$^{+/-}$ cells, respectively, (bottom). $n = 3$ independent experiments. Error bars represent SEM. *P*-values were determined by Student's *t*-test. *$P < 0.05$. **$P < 0.01$, ***$P < 0.001$

(Supplementary Fig. 6) and an increased percentage of benzidine-positive cells (Fig. 6i). Additionally, since ALAS2 catalyzes the production of 5-amino-levulinic acid (5-ALA) in the heme biosynthetic process[7], we predicted that the addition of 5-ALA could supplement the missing metabolite, thus bypassing the heme biosynthetic defect. To test this hypothesis, we infected lentiviral-mediated UCA1-targeting shRNA (#2) into primary erythroid cells differentiated for 4 days and then treated the cells with 5-ALA (300 μM) on day 6. At day 14, we found that 5-ALA partially rescued globin synthesis by increasing the percentage of benzidine-positive cells in UCA1-downregulated cells (1.7-fold induction, $P = 0.01$ determined by Student's $t$-test) (Fig. 6j). One explanation for the partial rescue is other UCA1-regulated targets that are not rescued by heme administration may also be important. Of potential relevance to this point is the fact that GSEA revealed that UCA1 is linked to multiple cellular and biological processes in erythroid cells (Fig. 4b). Taken together, ALAS2 overexpression or 5-ALA treatment partially rescued phenotypes induced by UCA1 downregulation in AraC-induced K562 cells or in primary erythroid cells.

To investigate the role of UCA1 in this protein–RNA complex, we conducted PTBP1 RIP assays in the $UCA1^{+/-}$ cells after ALAS2 overexpression. Although $ALAS2$ mRNA expression increased approximately 100-fold (Fig. 6g), its enrichment was still significantly lower in $UCA1^{+/-}$ cells in comparison with WT control cells (Fig. 6k). These results suggest that UCA1 specifically confers PTBP1-mediated $ALAS2$ mRNA stability by acting as a protein:RNA scaffold.

**GATA1 increases UCA1 expression during erythroid maturation**. To analyze how UCA1 expression might be transcriptionally regulated in erythroid cells, we predicted potential transcription factor motifs lying within 2 kb of the UCA1 TSS using JASPAR[46]. Two GATA1 motifs reside in the UCA1 proximal promoter at -262 and -1977 nt 5′ to the TSS, respectively (Fig. 7a). We tested whether GATA1 occupies these sites in erythroid cells on day 8 of differentiation from cord blood CD34[+] cells. ChIP assay, followed by qPCR, revealed GATA1 occupancy at the UCA1 promoter GATA sites (Fig. 7b), consistent with prior GATA1 ChIP-seq in primary erythroblasts derived from adult peripheral blood CD34[+] cells (Supplementary Fig. 7A)[47]. We also analyzed the expression of GATA1, UCA1, PTBP1, and ALAS2 during erythroid differentiation (Supplementary Fig. 7B).

To explore whether GATA1 occupancy at these motifs reflects its capacity to regulate transcription of the UCA1 promoter, we utilized a luciferase reporter assay in 293T cells with or without GATA1 expression. GATA1 significantly induced UCA1 promoter activity (Supplementary Fig. 7C, D). Deletion of the GATA-binding site(s) abrogated GATA1-mediated activation of the reporter-driven UCA1 promoter (Supplementary Fig. 7C, D). Reporter gene analysis in K562 cells with endogenous GATA1 revealed similar results (Fig. 7c).

To further explore the significance of the GATA1 occupancy, we infected CD34[+] cells at day 4 during erythroid differentiation with shGATA1 lentiviruses. GATA1 mRNA and protein expression was significantly downregulated (Fig. 7d, e), resulting in impaired erythroid differentiation, based on FACS analysis (Fig. 7f). UCA1, ALAS2, and PTBP1 expression were significantly reduced after GATA1 knockdown (Fig. 7g–i). Thus, GATA1 occupies the UCA1 promoter and regulates UCA1 expression, suggesting a potential direct transcriptional regulatory mechanism.

## Discussion
In addition to functioning as an enzyme cofactor and hemoglobin component in erythroid cells, heme exerts important activities to control erythropoiesis[6]. Heme deficiency causes erythroid diseases, such as sideroblastic anemia or erythropoietic porphyria. Despite firm links between heme and erythrocyte biology/pathology[1,2], many unanswered questions exist regarding heme regulatory mechanisms. We elucidated a lncRNA-dependent mechanism that controls heme biosynthesis in erythroid cells. UCA1 deficiency inhibited erythroid differentiation at the proerythroblast stage, which involved deregulation of $ALAS2$ mRNA stability and, therefore, impaired heme biosynthesis. Mechanistic analyses revealed that UCA1 functioned as an RNA scaffold to recruit PTBP1 to $ALAS2$ mRNA, and this complex conferred $ALAS2$ mRNA stability. This post-transcriptional mechanism provides a new dimension into the regulation of expression of the rate-limiting enzyme of heme biosynthesis. Finally, we demonstrated that GATA1 occupied the UCA1 promoter, suggesting a mechanism of transcriptional activation and establishing a novel component in the circuitry by which GATA1 regulates heme biosynthesis.

Another key finding of our study is the identification of PTBP1 as a cofactor in the UCA1-mediated ALAS2 regulatory circuit. PTBP1 functions in T-cell activation[48], hepatitis C virus (HCV) replication and infection[49], insulin secretion[50], and apoptosis[51]. Mechanistically, PTBP1 modulates mRNA metabolic steps including splicing, stability, and translation[44]. Our results indicate that UCA1 interaction with PTBP1 in erythroid cells post-transcriptionally affected $ALAS2$ mRNA stability. UCA1 interaction with PTBP1 had been described in breast cancer cells, where the interaction induced IRES-mediated translation initiation of p27 mRNA to promote cancer cell proliferation[52]. This interaction involved a UCA1 segment containing a conserved PTBP1 RNA-binding motif[52].

UCA1- and PTBP1-regulated heme metabolism involves the control of ALAS2 expression. $Alas2^{-/-}$ mutants are embryonic lethal in mice, as they cause severe anemia due to heme deficiency, which arrests erythroid differentiation at the proerythroblast stage[10,13]. Our results indicate that UCA1 or PTBP1 downregulation impaired heme biosynthesis and arrested erythroid differentiation at the proerythroblast stage. Thus, the phenotypes resulting from UCA1 and PTBP1 downregulation resemble those of ALAS2 deficiency in vivo.

ALAS2 expression in erythroid cells is regulated at transcriptional, translational, and post-translational levels. GATA1 occupies ALAS2 promoter and intronic GATA motifs and forms an enhancer loop that activates transcription[5,11–13]. Translationally, IRPs bind to IREs in $ALAS2$ mRNA 5′-untranslated region to prevent translation under conditions of iron deficiency[14]. Post-translationally, ALAS2 is ubiquitinated and degraded in the presence of excessive heme[53,54]. Our study uncovered an additional layer of regulation in human erythroid cells, in which a non-coding RNA regulates the post-transcriptional control of $ALAS2$ mRNA stability.

LncRNAs, like mRNAs, are transcriptionally regulated[40]. We demonstrated that GATA1 induced UCA1 expression and is, therefore, a potential initiator of the UCA1-mediated $ALAS2$ mRNA stability regulatory circuit. GATA1 modulates heme biosynthesis by activating transcription of multiple genes encoding heme biosynthesis enzymes[55], and our study illustrates a new mechanism by which GATA1 regulates heme biosynthesis in primates.

Defective heme biosynthesis in erythroid cells causes sideroblastic anemia or erythropoietic porphyria[56]. X-linked sideroblastic anemia (XLSA, OMIM#300751), caused by $ALAS2$ mutations, is the most common form of sideroblastic anemia[57]. Approximately 60% of XLSA cases can be traced to pathogenic $ALAS2$ mutations, resulting from corruption of either coding exons or non-coding regulatory elements[13,58,59]. The pathogenic

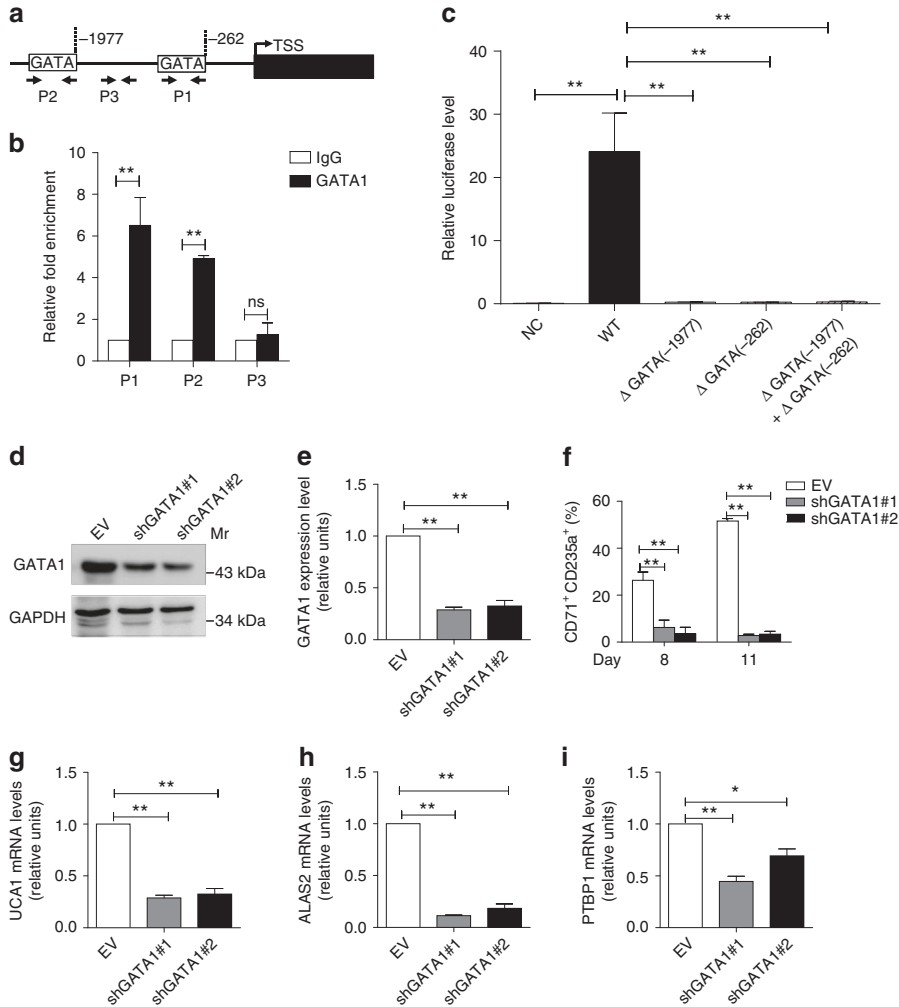

**Fig. 7** GATA1 regulates *UCA1* expression during erythroid maturation. **a** Schematic representation of GATA motifs in the *UCA1* promoter from -2 kb to the transcription start site (TSS). The white boxes depict GATA motifs, resided in the *UCA1* proximal promoter at -262 and -1977 nt 5′ to TSS. P1, P2, and P3 are the primer sets used for ChIP-qPCR assay. **b** ChIP-qPCR assay analyzed GATA1 occupancy at the *UCA1* promoter in primary erythroid cells after 8 days of differentiation. P3 was used as a negative control. **c** Luciferase report assay demonstrating GATA1 regulates *UCA1* promoter transcription in a transient transfection assay. The relative luciferase activity was examined after transfection of empty pGL3 luciferase vector (NC), pGL3 containing the wild-type *UCA1* promoter (WT), and pGL3 containing *UCA1* promoter with different mutated GATA-binding site(s) (ΔGATA(-1977), ΔGATA(-262), ΔGATA(-1977) + ΔGATA(-262)) in AraC-induced K562 cells. **d**, **e** qRT-PCR (**d**) and immunoblotting (**e**) and analysis of GATA1 expression at day 8 in differentiated primary erythroid cells with or without GATA1 depletion. GAPDH and *18S* rRNA were used as controls for WB and qRT-PCR, respectively. **f** The percentage of CD71$^+$CD235a$^+$ cells evaluated by flow-cytometric analysis on days 8 and 11 differentiated primary erythroid cells with or without GATA1 downregulation. **g–i** qRT-PCR analysis of the relative abundance of UCA1 (**g**), ALAS2 (**h**), and *PTBP1* (**i**) mRNA in differentiated primary erythroid cells at day 8 with or without GATA1 depletion. n = 3 independent experiments. Error bars represent SEM. *P*-values were determined by Student's *t*-test. *$P < 0.05$, **$P < 0.01$

basis of the remaining 40% of XLSA cases remains unclear and, therefore, may be caused by *ALAS2* mutations that have not been detected or indirect mechanisms. It will be interesting to evaluate the UCA1/PTBP1-dependent mechanism described herein vis-à-vis in human heme biosynthetic disorders, such as XLSA with unknown etiology.

Besides heme biosynthesis, UCA1 and PTBP1 downregulation was linked to additional cellular processes and signaling pathways in erythroid cells, such as G2M checkpoint, p53 pathway, and hypoxia. These results suggest that the UCA1 and PTBP1 interaction is likely to control heme biosynthesis as well as other important processes or pathways in erythroid cells. This may explain the partial rescue of erythroid differentiation upon ALAS2 overexpression.

Interestingly, although the UCA1 primary sequence is not significantly conserved in mammals, this lncRNA is crucial for human erythroid differentiation. It is unclear whether non-conserved lncRNAs commonly exert important functions. A systematic comparison of matched cell types generated from primary human and mouse tissues revealed ~80% of lncRNAs in primary human erythroid cells, including abundantly expressed lncRNAs, were not detected in mice (or vice versa)[21]. Furthermore, other species-specific, less conserved lncRNAs also appear to be functionally important. For example, Braveheart, a lncRNA critical for cardiovascular development in mice, is not expressed in rats or humans;[60] LncND, which is expressed specifically in primates but not in other mammals, controls brain development[61]. Clearly, primary sequence conservation does not invariably predict functional importance.

In summary, the present study provides evidence that lncRNA UCA1 controls heme biosynthesis. It will be particularly instructive to further dissect mechanisms governing UCA1/

PTBP1 function in this specific physiological context and to establish whether this system remains intact or exhibits vulnerabilities that may underlie hematologic pathologies.

## Methods

**Ex vivo differentiation of human CD34+ cells.** Human umbilical cord blood was obtained from Tianjin Obstetric Central Hospital (Tianjin, China) after informed consent approved by the Ethical Committee on Medical Research at Institute of Hematology. After isolation by HISTOPAQUE® (Sigma, Cat# 10771) density gradient centrifugation, mononuclear cells (MNC) were passed through CD34 microbeads (Miltenyi Biotec, Cat# 130-046-703) for purification.

The purified CD34+ cells were differentiated ex vivo toward the erythroid lineage[37]. In detail, CD34+ cells were cultured in the basic Iscove's modified Dulbecco's medium (IMDM) (Sigma, Cat# 13390) supplemented with 20% BIT (v/v) (StemCell Technologies, Cat# 9500), 40 μg per ml inositol (Sigma, Cat# I5125), 10 μg per ml folic acid (Sigma, Cat# F7876), 160 μM monothioglycerol (Sigma, Cat# M6145), 90 ng per ml ferrous nitrate (Sigma, Cat# 8508), 900 ng per ml ferrous sulfate (Sigma, Cat# F8633), 2 mM L-glutamine (Gibco, Cat# 25030) and 1% penicillin and streptomycin (v/v). From days 0 to 8, 5 ng per ml of recombinant human interleukin-3 (IL-3) (Sigma, Cat# I1646), 100 ng per ml recombinant human stem cell factor (SCF) (Prospec, Cat# CYT-255), 1 μM hydrocortisone (Sigma, Cat# H2270) and 3 U per ml of recombinant human Erythropoetin alfa (EPO) (ProTech, Cat# 100-64) was supplemented to the culture medium. From days 8 through 14, SCF and EPO were included in the culture medium, and only EPO was included during days 14 to 18. Cells were maintained at a density between $5 \times 10^4$ and $2 \times 10^6$ cells per ml, with medium changes every 3 to 4 days. Cells were incubated at 37 C with 5% $CO_2$. Cell morphology was analyzed by Wright-Giemsa staining (Baso, Cat# BA-4017). Hemoglobin was assessed by benzidine staining[62]. Briefly, $2 \times 10^5$ cells in 50 μl culture media were placed in 96-well plate and stained with 10 μl freshly prepared benzidine staining solution (150 μl 10% acetic acid (v/v), 20 μl benzidine solution and 10 μl 30% $H_2O_2$) at room temperature (RT) for 1 min in the dark. The gray and/or black cells were scored as benzidine-positive.

**HUDEP-2, 293T, and K562 cell culture.** Human umbilical cord blood CD34+-derived erythroid progenitor cells (HUDEP-2), obtained from RIKEN BioResource Research Center, were cultured in StemSpan SFEM® medium (Stem Cell Technologies, Cat#09650), in the presence of SCF (50 ng per ml), EPO (3 IU per ml), doxycycline (1 μg per ml) (Sigma, Cat# 9891), dexamethasone ($10^{-6}$ M) (Sigma, Cat# 9891D2915) and 1% penicillin and streptomycin. In all, 293T and K562 cells were obtained directly from the American Type Culture Collection (ATCC) and cultured according to ATCC recommendations. 293T cells were maintained in Dulbecco's modified Eagle's medium (DMEM) (Gibco, Cat# c11330500BT) with 10% fetal bovine serum (FBS) (Gibco, Cat# 16000-044) and 1% penicillin and streptomycin. Human erythroleukemia K562 cells were maintained in RPMI 1640 (Gibco, Cat# c11875500BT) supplemented with 10% FBS and 1% penicillin and streptomycin. Erythroid differentiation of K562 cells was induced with 20 nM AraC (Sigma, Cat# C6645) for 48 h.

**Measurement of cellular size and nuclear size.** To measure nuclear size, cells were isolated on slides by cytospin, fixed in 3.7% polyformaldehyde for 20 min and stained with 10 μg per ml Hoechst 33342. The slides after Wright-Giemsa staining were used to measure cell size. Cell and nuclear diameter was measured using Nano measurer software. Cell clumps and debris were excluded from analysis by setting appropriate cutoffs after visual inspection of the images. For each sample, ~100 cells were analyzed.

**Fluorescence-activated cell sorting (FACS) analysis.** For FACS analysis the cells were collected, washed, and resuspended in ice-cold phosphate-buffered saline (PBS) with 2% FBS. For each assay, $1 \times 10^5$ cells in 100 ml PBS-FBS were stained with 0.2 μg PE-conjugated anti-CD235a (eBioscience, Cat# 4285151) and 0.06 μg APC-conjugated anti-CD71 (Invitrogen, Cat# 4331118) antibody at 4 °C for 30 min in the dark. Cells were then washed with cold PBS, resuspended in 300 ml ice-cold PBS-FBS, and subjected to flow-cytometric analysis on a FACS LSRII flow cytometer (BD Biosciences). Data were analyzed with FlowJo (version 7.6.1).

**Plasmid construction, lentivirus production, and infection.** The short hairpin RNA (shRNA) sequences were cloned into lentiviral shRNA expression vector pLKO.1 empty vector (EV) with puromycin resistance. To overexpress ALAS2 and GATA1, the respective cDNAs were amplified from K562 cells and cloned into piggybac transposon expression vector (a generous gift from Wang Y. at Peking University in China). Lentiviruses were packaged using ViraPower Lentiviral Packaging System (Invitrogen). For infection, K562 cells or primary erythroid cells at day 4 of differentiation were incubated with lentiviruses for 12 h before washing the excess virus. Forty-eight hours after infection, the cells were selected with 1 μg per ml puromycin until the end of the culture period. The primers used for shRNA sequences or overexpression of ALAS2 or GATA1 are listed in Supplementary Table 1.

**RNA extraction and qRT-PCR.** Total RNA was extracted using TRIzol (Invitrogen, Cat# 15596026), and cDNA was synthesized using a reverse transcription system from Promega (Cat# 0000223677). Quantitative real-time PCR (qRT-PCR) was performed using the SYBR Green PCR kit (Applied Biosystems, Cat# A25742). Primer sequences used for qRT-PCR are shown in Supplementary Table 2.

**Western blotting (WB).** Cells were lysed in Laemmli sample buffer (BioRad, Cat# 161-0737) and subjected to sodium dodecyl sulfate polyacrylamide gel electrophoresis (SDS-PAGE). Proteins were transferred to a nitrocellulose membrane and incubated with antibodies to PTBP1(1:1000, Abcam, Cat# ab5642), GAPDH (1:5000, Cell Signaling Technology, Cat#5174), ALAS2 (1:1000, Abcam, Cat# ab184964) or GATA1 (1:1000, Abcam, Cat# ab11852). Horseradish peroxidase-conjugated secondary antibodies were used. Signals were detected using an ECL kit (Invitrogen, Cat# WP20005). Unprocessed images of all western blots are shown in Supplementary Fig. 8.

**RNA fluorescence in situ hybridization (RNA-FISH).** RNA in situ hybridization was performed using RNAscope® Multiplex Fluorescent Reagent Kit V2 (ACD, Cat# 323100)[41] according to the manufacturer's instructions. Mouse erythroleukemia cells (MELs) and primary erythroid cells at day 8 of differentiation were fixed in 10% polyformaldehyde at 37 °C for 1 h and collected by cytospin ($1 \times 10^6$ cells per ml). The air-dried slides were sequentially immersed in 50, 70, 100% ethanol for 5 min each and incubated at 37 °C for 30 min to dry. After adding 100 μl protease III, slides were placed in a pre-warmed, humidified tray and then incubated in HybEZ™ Oven at 40 °C for 30 min to penetrate the membrane. For hybridization, slides were incubated with 20 sets of *UCA1* DNA probes (ACD, Cat# 417521), peptidylprolyl isomerase B (*PPIB*) probes (positive control) (ACD, Cat# 313911), and bacterial gene *dapB* probes (negative control) (ACD, Cat# 310043), respectively, for 2 h at 40 °C in HybEZ™ Oven. To amplify the signals, slides were incubated with the amplification solution I, II, and III for 30, 30, and 15 min, respectively, at 40 °C in a humidified chamber. After washing, slides were further amplified with HRP-C1 solution for 15 min at 40 °C in a humidified chamber. Finally, slides were incubated with 200 μl TSA® Plus Fluorescein (1:1500 dilution) for 30 min at 40 °C in a humidified chamber followed by 2–4 drops of DAPI for 30 s at RT.

We utilized a confocal microscope (UltraVIEW Vox, PerkinElmer) to quantify the RNA-FISH data. Samples were imaged on a spinning disk confocal microscope equipped with a 100 × objective and 405 and 488 nm lasers. Representative cells were acquired by the same devices with Z stacks. Quantitative analyses and reconstruction into three-dimensional images were conducted using Volocity (version 6.0, PerkinElmer).

**Subcellular fractionation.** In all, $2 \times 10^7$ cells were harvested and washed with ice-cold PBS before resuspension in ice-cold cytoplasmic lysis buffer (0.15% NP-40, 10 mM Tris pH 7.5, 150 mM NaCl) for 5 min on ice[63]. Lysates were transferred onto ice-cold sucrose buffer (10 mM Tris pH 7.5, 150 mM NaCl, 24% sucrose w/v) and spun at $13,000 \times g$ for 10 min at 4 °C[63]. The supernatant (~700 μl) was collected as the cytoplasmic fraction.

**In vitro RNA pull-down assay.** The RNA pull-down assay was modified from previous studies[64,65]. Substrate RNAs for bead immobilization were in vitro transcribed in a 200 μl reaction mix containing 1 μg of template DNA of *UCA1* or *ALAS2*, 40 μl 5x transcription buffer, 8 μl NTP mixture (Invitrogen, Cat# R0481), 5 μl RNase inhibitor (Promega, Cat# N2511) and 10 μl T7 RNA polymerase (Invitrogen, Cat# EP0111) at 37 °C for 12 h, followed by RNA purification with 6% denaturing urea polyacrylamide gel. These in vitro transcribed RNAs contained a GGC sequence at their 5′ ends to improve transcription efficiency. RNAs were oxidized in a reaction mixture containing 100 mM sodium acetate and 5 mM sodium periodate (Sigma, Cat# 71859) for 1 h in the dark at room temperature (RT). Then RNAs were ethanol-precipitated and resuspended in 500 μl 0.1 M sodium acetate. Meanwhile, 200 μl of adipic acid dihydrazide agarose bead (Sigma, Cat# A0802) slurry was washed four times with 0.1 M sodium acetate. The beads, resuspended in 300 μl 0.1 M sodium acetate, were then incubated with the sodium periodate-treated RNAs on a rotator for 12 h at 4 °C. After washing the RNA-bead complexes with 2 M NaCl and NP-40 buffer (150 mM NaCl, 1.0% NP-40, 50 mM Tris pH 8.0) three times, the RNA-bead complexes were incubated with the cytoplasmic extracts at 30 °C for 20 min. Beads were washed, in order, with low salt NP-40 buffer (150 mM NaCl, 1.0% NP-40, 50 mM Tris pH 8.0) and high salt NP-40 buffer (350 mM NaCl, 1.0% NP-40, 50 mM Tris pH 8.0) three times. After the final centrifugation, proteins bound to the immobilized RNAs were eluted by adding SDS buffer and boiled for 10 min. The recovered proteins were analyzed by WB or resolved by gradient gel electrophoresis for identification by mass spectrometry. Primer sequences used for amplification of the template DNA are shown in Supplementary Table 3.

**Mass spectrometry (MS) analysis.** Proteins eluted from adipic acid dihydrazide agarose beads by in vitro RNA pull-down assay were conducted after gel electrophoresis on SDS-PAGE. After staining with Coomassie blue, the gel was cut into slices and subjected to in-gel trypsinization overnight at 37 °C. The trypsinized

peptides were directly sprayed into the nano-ESI source of the mass spectrometer (Thermo Scientific). Mass spectra were acquired in continuum mode; fragmentation of the peptides was performed in data-dependent mode at a resolution of 120,000 followed by CID (Collision Induced Dissociation) MS/MS scans on the 15 most abundant ions in the initial MS scan.

**RNA immunoprecipitation (RIP).** In all, $2 \times 10^7$ cells were harvested and cross-linked with 1% formaldehyde for 10 min at RT and quenched with 0.125 M glycine for 5 min. Cells were washed and resuspended in 400 μl lysis buffer (50 mM Tris pH 7.4, 150 mM NaCl, 1% Triton X-100, 5% glycerol, supplemented with 1 mM DTT, 1 mM phenylmethylsulfonyl fluoride (PMSF), protein inhibitors cocktail and 400 U per ml RNase inhibitor) before rotating for 40 min at 4 °C. The protein-G Dynabeads (Invitrogen, Cat#10004D) were washed with dilution buffer (50 mM Tris pH 7.4, 150 mM NaCl, 1 mM EDTA, 0.1% Triton X-100) and blocked with 50 ng yeast total RNA for 1 h at 4 °C before use. Four micrograms of anti-PTBT1 or goat IgG (Solabio, Cat# SP038) antibody was incubated with the protein-G Dynabeads (Invitrogen, Cat#00585833) and rotated at RT for 30 min before incubation with whole-cell lysates overnight at 4 °C. After 12 h, beads were washed five times with 0.5 ml IP200 buffer (20 mM Tris pH 7.4, 200 mM NaCl, 1 mM EDTA, 0.3 Triton X-100, 5% glycerol). For elution, RNA–antibody complexes were digested with 80 μg Proteinase K in 200 μl of digestion buffer (50 mM Tris pH 7.4, 10 mM EDTA, 50 mM NaCl, 0.5% SDS) and vortexed at 37 °C for 1 h and at 55 °C for 30 min in order. Finally, RNAs were extracted with TRIzol and detected by qRT-PCR[66]. To avoid detecting the interaction of PTBP1 and nuclear RNAs, the primers for all detected genes flanked exon–exon junctions.

**In vivo RNA–RNA pull-down assay with biotinylated DNA probes.** To perform the pull-down assay[67,68], we synthesized biotinylated DNA probes complementary to UCA1 or ALAS2 RNA and incubated them with streptavidin-coated agarose beads (Invitrogen, Cat# S951) at 4 °C for 10 h to generate probe-coated agarose beads. Whole-cell lysates of HUDEP-2 or AraC-induced K562 cells were precleared with 50 μl avidin-agarose beads (Pierce, Cat# 20219) at 4 °C for 1 h to remove endogenous biotin and incubated with probe-coated beads sequentially at RT for 3 h and 4 °C overnight. After washing with the wash buffer (0.5 M NaCl, 20 mM Tris pH 7.5, 1 mM EDTA), the RNA complexes bound to the beads were eluted and extracted for qRT-PCR. The RNA pull-down probes are listed in Supplementary Table 4.

**Generation of UCA1 compound heterozygous mutant cell line.** Single-guide RNAs (sgRNA) targeting human UCA1 gene were designed in http://crispr.mit.edu/. Oligonucleotides were cloned into the lentiviral vector pL-CRISPR.EFS.GFP (Addgene) and pLKO5.sgRNA.EFS.tRFP (Addgene), respectively. The most effective pair of sgRNAs was used in the following experiments. After 48 h of lentivirus infection in K562 cells, the GFP and RFP double-positive cells were sorted as single cells using an Aria III instrument (BD Biosciences). Colonies emerging from every single cell were expanded and one UCA1 compound heterozygous mutant cell line was identified using genotyping and Sanger sequencing. The sgRNA sequences are: 5′-CACAGGGUCGATTGAUCAGU-3′ and 5′-UCTATACGGACACG CGUGAC-3′.

**Chromatin immunoprecipitation (ChIP).** A total of $1 \times 10^7$ erythroid cells differentiated for 8 days were harvested, fixed with 1% formaldehyde for 10 min, and quenched with 0.125 M glycine for 5 min. After cells were lysed in cell lysis buffer (5 mM PIPES pH 8.0, 85 mM KCl, 1% Igepal) supplemented with protease inhibitors (aprotinin 10 μg per ml, leupeptin 10 μg per ml, and 1 mM PMSF) on ice for 15 min, nuclei were collected and lysed in nuclei lysis buffer (50 mM Tris, pH 8.0, 10 mM EDTA, 1% SDS) containing the protease inhibitors for 30 min. Then the DNA was sheared into 200 to 500 bp by sonication. The DNA lysate in IP dilution buffer (50 mM Tris, pH 7.4, 150 mM NaCl, 1% Igepal, 0.25% deoxycholic acid, 1 mM EDTA) containing protease inhibitors was incubated with 2 μg GATA1 antibody (Abcam, Cat# ab11852) or IgG (Millipore, Cat# 2778875) for 12–16 h on a rotating platform at 4 °C, followed by the addition of 20 μl protein-G agarose beads for 2 h by rotation. Then, the antibody/chromatin complexes were eluted by vortexing in elution buffer (50 mM NaHCO₃, 1% SDS) for 30 min at RT. Final concentration of 0.54 M NaCl was added to the immunoprecipitates and heated 67 °C in water bath for 4 h to reverse the cross-linking. Finally, 1 μl of RNase A was added and incubated at 37 °C for 20 min. The DNA was purified by using a QIAquick PCR purification kit (Qiagen, Cat# 28104) and then subjected to qRT-PCR. All the primers used for ChIP-qPCR are listed in Supplementary Table 2.

**Luciferase reporter assay.** The fragments of UCA1 promoter (-2kb–0 bp) containing the GATA1-binding sites and corresponding GATA site(s) deletion (as shown in Fig. 7a) were inserted into pGL3 basic constructs. 293T cells were co-transfected with 0.5 μg reporter construct, 0.02 μg pRL-TK vector, and 0.5 μg piggybac-GATA1 or empty control (PB) per well using Lipofectamine 3000 (Invitrogen, Cat# L3000-015). K562 cells were co-transfected with 0.5 μg reporter construct and 0.02 μg pRL-TK vector to further confirm the regulation of endogenous GATA1 on UCA1 transcription. After 12 h of transfection, we replaced transfection medium with complete culture medium. After 48 h culture, the cells

were lysed with passive lysis buffer (Promega, Cat# E1910), and reporter gene expression was assessed using the Dual Luciferase reporter assay system (Promega, Cat# E1910). All transfection assays were carried out in triplicate.

**RNA-Seq.** Total RNA from two biological replicates was extracted using TRIzol (Invitrogen, Cat# 15596026). Library construction and data processing were performed by Novogene (Beijing, China). The RNA integrity was measured by Agilent 2100. After RNA quality check, rRNAs were removed with Epicenter Ribo-Zero™ Kit (Epicenter). The remaining RNAs were fragmented (250–300 bp) and reverse transcribed using random hexamers. Followed by purification, terminal repair, polyadenylation, adapter ligation, size selection, and degradation of second-strand U-contained cDNA, the strand-specific cDNA library was generated. The cDNA was sequenced with an Illumina HiSeq platform, and 150 bp paired-end reads were generated.

**Alignment and quantification of expression.** The human genome sequence (GRCh38/hg38) and annotation GTF file (GENCODE version 27) were obtained from https://www.gencodegenes.org/. Clean reads were aligned using HISAT2[69]. We applied the parameter (--rna-strandness RF) to distinguish the first-strand-specific cDNA library. The aligned reads were then used to quantify gene and transcript isoform expression by the StringTie[69] software with default parameters. We applied the parameter (--rf) to distinguish the first-strand-specific cDNA library.

**Identification of differentially expressed genes (DEGs).** All gene expression data with average FPKM > 0.1 were used to identify DEGs. DEGs were identified using DESeq2[70] software with default parameters. The P-values were adjusted using the Benjamini and Hochberg method and the threshold was set to $P_{adj} < 0.05$.

**Functional annotation of DEGs.** The hallmark enrichment analysis was conducted using hallmark database from MSigDB (Molecular Signatures Database, version 6.2) and the GO enrichment analysis was carried on using the GOTERM_BP_-DIRECT database from DAVID (https://david.ncifcrf.gov/home.jsp). The DEGs were converted to Entrez gene ids first and then performed the enrichment analysis. The P-values were adjusted using the Benjamini and Hochberg method and the threshold for significantly enrichment was $P_{adj} < 0.05$.

**Gene sets enrichment analysis (GSEA).** GSEA was conducted, and the plots was generated by GSEA software (version 3.0) from the Broad Institute (http://software.broadinstitute.org/gsea/index.jsp). Heme metabolism gene set was derived from MSigDB (HALLMARK_HEME_METABOLISM). We input the gene expression matrix with average FPKM > 0.1 from all samples with replicates. The numbers of permutations were set to 1000. All P-values were corrected for multiple testing method and the threshold for significantly enrichment is P < 0.05 and FDR < 0.25.

**Heatmap generation.** Heatmaps were generated using the R package pheatmap (version 1.0.8). The input values were log transformed FPKM or fold-change. Heatmaps were centered and scaled in the row direction. Euclidean distance and complete method were used for Hierarchical Clustering.

**Statistical analysis.** Student's t-test (two-tailed) was conducted to analyze the data. Statistical significance was set at P < 0.05.

## Data availability

RNA-seq data have been deposited in NCBI's Gene Expression Omnibus under the accession GSE106567. All data that support the findings of this study are available from the corresponding authors upon request.

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

## Acknowledgements

This work was supported by the National Key Research and Development Program of China Stem Cell and Translational Research (2016YFA0102300, 2017YFA0103100 and 2017YFA0103102 to L.S.); CAMS Initiative for Innovative Medicine (2016-I2M-3-002 to L.S., 2016-I2M-1-018 to J.L. and 2017-I2M-1-015 to J.G.); National Natural Science Foundation of China (31471291 and 81870089 to L.S. and 81700105 to J.L.); CAMS Medical Epigenetics Research Center (2018PT31033); Tianjin Natural Science Foundation (15JCYBJC54500 to L.S.); and NIH (DK50107 to E.H.B. and HL117658 to J.D.E.). We thank members of the Jiaxi Zhou and Lihong Shi laboratory, Dr. Ma, Dr. Yu, Dr. Chen and Dr. An for their insightful discussion during the course of this work. We thank Dr. Xu for providing antibody for this study.

## Author contributions

J.L., Y.L., J.T., J.G., Q.G., L.Z., and B.W. performed research and analyzed the data. R. K., Y.N. prepared and helped for culture of HUDEP-2 Cells. H.Z., H.W., E.J. analyzed the data. O.T., E.H.B., J.D.E. analyzed the data and edited the paper. L.S. and J.Z. designed experiments, analyzed the data, and wrote the paper.

## Additional information

**Competing interests:** The authors declare no competing interests.

