## [Peer Review File · Nature Communications]

Reviewers' comments:

Reviewer #1 (Remarks to the Author):

This is an interesting manuscript that describes the role of UCA1 in erythroid cells. The data is pretty extensive and almost all experiments are convincing. The major issue with the paper is the specificity of the effect of UCA1 on ALAS2: Figure 4D suggests that UCA1 and PTBP1 depletion affects very large number of genes and that the heme synthesis pathway is not the major target of these genes. These does not mean that the conclusion of the manuscript are incorrect but that the authors have cherry picked one of many effect of UCA1 which might affect mRNA stability or other aspects of the regulation of most gene in erythroid cells.

Similarly, even within the heme biosynthesis pathway, the authors focused exclusively on ALAS2 which leaves the impression that this mechanism might only affect ALAS2. Author should discussed ow specific they believe the mechanism to be. The binding site for PTBP1 is very small and most likely present in many mRNA. Is there any evidence that the proposed mechanism is specific for ALAS2 ? Is once knocks-out RNA polymerase II one might see an effect on ALAS2 but reporting that POIII affect ALAS2 regulation might not be that meaningful.

Figure 2:

Expression of the genes involved in heme metabolism that are believed to be regulated by UCA1 should be shown, rather than expression of relatively irrelevant genes. Differential regulation should be demonstrated by Q-PCR.

Figure 6: the data is not really convincing that GATA1 regulate UCA1. A crispr in K562 deletion of the GATA1 would be more convincing becauseu the assay that were performed could be associate to indirect effects.

Description of the RNA-seq experiments and of the bioinformatics processing is not detailed enough

Reviewer #2 (Remarks to the Author):

The manuscript by Liu and colleagues details studies on the role of UCA1 lncRNA in erythroid cell development. The authors either primary umbilical cord-derived erythroid progenitor cell or the erythroid leukemia cell line K562 cells to first, screen for lncRNAs showing differential expression at the proerythroblast stage compared to earlier progenitor stages, and identify UCA1 lncRNA as one of the most differentially expressed non-coding transcript during this transition. Next, the authors use a knock down approach to show a requirement for the expression of this lncRNA in erythroid differentiation. The authors next attempt to identify the interacting factors that mediate the function of this lncRNA and the rest of the manuscript is devoted to various knock down, overexpression and affinity purification experiments aiming to mechanistically elucidate the mode of action of UCA1 in erythroid differentiation. While study of the role of lncRNAs in differentiation of blood cells is certainly of great value and the initial observations of the authors are very interesting, unfortunately several major shortcomings strongly affect the impact of this study. Further, the manuscript is written in a highly minimalistic way, to the point that some experiments are impossible to evaluate, and it's frequently necessary to combine the main text, materials and methods and the figure legends to barely get a grasp of what the experiment entails. The data are over-interpreted and alternative hypotheses have not been considered. Many key controls are missing and common experimental procedures aiming at reducing artifactual data have been ignored. Some of these issues are detailed below, along with some additional shortcomings.

1) The majority of the studies beyond the initial characterization detailed above has been carried out in K562 cells, an erythroleukemia cell line, and neither of the findings have been validated in primary cells. Thus, the majority of the manuscript is focused on the mechanism of erythroid differentiation in a cancerous cell line, which may or may not reflect the situation in primary cells. This is particularly a concern when studying the function of lncRNAs, which have an expression pattern and mechanism of function which is highly cell type and state-dependent.

2) Several key studies, including lncRNA pull down and overexpression studies, have been performed using highly artificial systems and outdated methodologies. The use of in vitro-transcribed RNAs for pull down assays, which would have been acceptable 6-7 years ago, is currently thought of as highly error-prone, as all RNAs fold and associate with proteins in a co-transcriptional manner and thus, such in vitro-transcribed RNAs are frequently both misfolded and unable to properly associate with their native set of interacting proteins. Several manuscripts detailing much more reliable methodologies using pull down of endogenous lncRNAs have been published in recent years and have become the standard procedure for defining the interactome of an RNA molecule for the last several years. The authors should use these much more reliable technologies to properly identify the interactome of UCA1 in primary cells.

3) Another example of the use of an outdated methodology and lack of appropriate controls is the in situ hybridization studies shown in Fig. 3A. For the last several years, it has become standard procedure to use multiple probes or sets of probes, to ensure that the observed signal is not the artifact of an off-target interaction of a particular probe, along with a control, non-targeting probe. As presently performed, it's not possible to rule out such an off-target effect, or even rule out that the observed signal is not an artifact of staining.

4) Have the RIP assays detailed on top of page 12 performed using whole cell extract or only cytoplasmic extracts? PTBP1 is mostly nucleoplasmic, and thus, the pull down from total cellular extracts may mostly yield information about the association of PTBP1 with the immature nuclear RNAs. Have the authors considered this possibility when performing RT-qPCR to detect UCA1, which is a spliced message? Have they used probes that flank each of the two exon-exon junctions in UCA1 to determine the share of immature versus mature UCA1 transcripts in the interaction they observe in RIP assays?

5) An example of overstating the data and ignoring alternative hypotheses is the interpretation of the PTBP1 knock down data on page 12. Knock down of proteins as critical to cellular RNA biogenesis and function as PTBP1 is expected to affect many RNAs, and thus disrupting many cellular processes, including differentiation. Until proven otherwise, such failure to differentiate phenotypes are not by any means due to a UCA1-mediated mechanism. Knock down of many other key factors in erythroid differentiation or general transcriptional and post-transcriptional regulatory mechanisms could result in failure to differentiate, with downstream effects identical to the ones described for UTA1 and PTBP1 knock downs. However, it does not mean that all such factors act through association with UTA1, as the authors state on page 12 and 13. In fact, this reviewer could not find a single firm, dependable data in the entire manuscript that proved that the association of UCA1 with PTBP1, assuming that it is a real interaction, has any role in erythroid differentiation beyond possibly supporting some aspect of the biogenesis of UCA1 itself which has not been shown nor investigated in this manuscript.

6) In Figs. 5A and 5B, the authors attempt to show, using RIP assays and the outdated RNA pull down assays detailed in point #2 above, that both UCA1 and ALAS2 interact with PTBP1, and partly based on these results, go ahead to conclude that UCA1, ALAS2 and PTBP1 must form a ternary complex. The authors don't seem to appreciate that PTBP1 and all other hnRNPs bind a very large number of cellular transcripts especially those that are spliced (in the case of PTBP1), and thus, observing PTBP1 interaction with cellular transcripts is not by any means unusual, especially if total cellular extracts are used in such assays.

7) The one experiment that could have shown some firm data supporting the claims of the authors is the one shown in Figs. 5C and D. However, there is literally NO information about how this has been done, and how the enrichment is measured. It should be mentioned that detection of such RNA-RNA interactions is by no means routine and thus, a very detailed experimental protocol should have been provided. In the absence of such information, this key data sadly remains uninterpretable.

8) The middle paragraph of page 14 is very confusing. The authors state that they make 3 probes, representing different lengths of ALAS2 mRNA. Are these antisense probes? They state that they were used for RNA pull down analyses followed by blotting for PTBP1. I am assuming that by probes they are referring to truncated versions of ALAS2 to see which one interacts with PTBP1 in an artificial system (in vitro-transcribed RNAs incubated in cellular extracts). While there is a strong signal in the immunoblot for PTBP1 in the lane corresponding to the full length ALAS2, all truncated versions, even those that the authors claim bind PTBP1, show either no signal or woefully weak signals that cannot be used to draw any conclusions about where the binding domain(s) of PTBP1 in ALAS2 mRNA is(are). Since PTBP1 is an RNA-binding protein with a very short binding motif that can be easily found in most RNAs longer than 500 nucleotides, such weak binding results should be treated as expectable artifacts. Also, in such very artificial systems, it is difficult to assess whether the various RNAs incubated in cellular extract demonstrate the same level of stability, thus further complicating the interpretation of such results. No information or even mention of the relative stability of the constructs in cellular extract is given. Thus, the failure to pull down PTBP1 could be simply due to total degradation of the truncated RNAs. Do the authors have any data to disprove this?

9) In Fig. 5H, the time point zero samples should not all be adjusted to one, but should be normalized to the time point zero value in the wild type sample. If the level is already significantly decreased in the knock down samples at time point zero, the interpretation and calculation of significance of the 24 hour values will be affected.

10) Another example of lack of appropriate controls and following standard procedures is that in most knock down studies in the manuscript, only one shRNA is used, creating the concern that what the authors present may be an off-target artifact. This is further exacerbated by the absence of a non-targeting shRNA as a control in these studies, which is needed to ensure that the observed effects are not the result of perturbations caused by the shRNA expression and biogenesis instead of their targeting function.

11) Can the authors explain why the reduction in the level of UCA1 RNA is so dramatic in a heterozygous knock out cell line?

12) Similar to the point #5 above, the interpretation of the results in Fig. S5D (detailed on page 15, 2nd paragraph) is problematic. Any gene that blocks erythroid differentiation will have the same effect as what is shown in Fig. S5D. Thus, this result does not indicate a specific relationship between UCA1 and ALAS2 RNAs.

13) Similarly, the results in Fig. 6B are incorrectly interpreted. The observed results are entirely expected based on the design of the experiment, since the total level of the ALAS2 RNA in the heterozygous knock out cell line is a lot less than it is in wild type cells. Further, the in vitro-transcribed RNAs differ from the endogenous mRNA in terms of their stability which further confounds the interpretation of the observed results. Instead of using such highly artificial systems, the authors should overexpress ALAS2 from a transgene in a living cell and repeat this experiment in multiple clones after measuring the level of overexpression while including all the needed controls.

14) Another example of the problems caused by the omission of key information in the text is Fig.

7C. From the text, it is not clear how this experiment is done at all. One has to read the figure legend and the materials and methods section and then only guess what each bar in the graph refer to. There is of course no information about how much GATA1 protein was overexpressed over the endogenous level, thus making it difficult to evaluate the significance of the results in the figure.

Minor points:

1) Page 9, 2nd paragraph, last line : the authors state that they have used a fold change threshold of 1.2 to identify differentially expressed lncRNAs. This seems an excessively low threshold which is not only unusual, but also not consistent with what is shown in Fig. 1F. Is this a typo, or another example of super-minimalism and omitting key information? Can the authors explain why they have chosen this threshold?

2) Key aspects of experiments must be included in the text, rather than buried in material and methods, which is frequently skipped by readers. For example, on page 11, last paragraph, it is essential to mention that the RNA pull down assays are performed using in vitro-transcribed RNAs incubated in extracts to ensure the readers can judge the merit of the shown data.

3) The results in Fig. 5H suggests that ALAS2 has a very long half life. Is this supported by other studies?

4) Page 15, two lines from the bottom. The authors need to describe what they mean by "quantitative" ChIP analysis.

Reviewer #3 (Remarks to the Author):

In this study, the authors identified UCA1 as a long non-coding RNA (lncRNA), which was upregulated specifically in the proerythroblast stage using ex vivo erythroid differentiation system of human cord blood CD34+ cells. UCA1 is a primate-specific lncRNA. Erythroid differentiation and hemoglobin synthesis were affected by knockdown of UCA1, indicating that UCA1 is required for erythroid differentiation. UCA1 localizes mainly in the cytoplasm. The authors identified PTBP1 that interacts with UCA1 by RNA pull-down assays with mass spectrometry analysis. Phenotype of PTBP1-knockdown cells resembles that of UCA1-knockdown cells, suggesting that UCA1 functions with PTBP1. The authors found that genes associated with heme metabolism were affected by both UCA1-knockdown and PTBP1-knockdown cells. In these genes, the authors focused on Alas2. They found that UCA1 lncRNA, PTBP1 protein and Alas2 mRNA formed a complex, which stabilized Alas2 mRNA. Finally, the authors found that GATA1 directly upregulates the UCA1 expression.

This study provides a novel insight for primate-specific lncRNA, which is required for erythroid differentiation. The paper is well written. To strength the authors' conclusion, the following points need to be considered;

1. The authors conclude that UCA1 induces erythroid differentiation by promoting heme synthesis, especially stabilizing Alas2 mRNA. To strength this conclusion, it may be appropriate for the authors to examine whether Alas2 overexpression rescues phenotype of UCA1-knockdown cells (erythroid differentiation and hemoglobin synthesis).

2. The level of UCA1 expression was the highest on day 8 during erythroid differentiation of CD34+ progenitor cell, while the induction of benzidine positive cells was further stimulated after

day 8. As the authors mention in Discussion, this reviewer considers multiple mechanisms confer to the function of ALAS2 during erythroid differentiation in a stage-specific manner. Therefore, this reviewer considers that the expression profiles of UCA1, PTBP1 and ALAS2 during erythroid differentiation should be shown in a panel, in addition to the expression profile of GATA1 that regulates both UCA1 and ALAS2 expression.

3. In Figure 6 and S5, the author established UCA1^{+/-} cells by CRISPR/Cas9 system. UCA1 RNA levels were dramatically decreased in UCA1^{+/-} cells, although one allele of UCA1 was intact. The author need to explain how they judge one UCA1 allele was intact.

4. The authors should describe how cellular and nuclear sizes were measured in Materials and Methods.

RESPONSE TO CRITIQUES

Manuscript reference number: **NCOMMS-17-30180A-Z**.

We thank the reviewers for instructive comments and recommendations that have contributed to the improvement of the manuscript. Below, we have articulated the point-by-point responses and revisions implemented, including new experimental data that addresses each of the reviewers' comments.

Reviewer comments:

Reviewer 1 (Remarks to the Author):

This is an interesting manuscript that describes the role of UCA1 in erythroid cells. The data is pretty extensive and almost all experiments are convincing. The major issue with the paper is the specificity of the effect of UCA1 on ALAS2:

- (1) Figure 4D suggests that UCA1 and PTBP1 depletion affects very large number of genes and that the heme synthesis pathway is not the major target of these genes. These does not mean that the conclusion of the manuscript is incorrect but that the authors have cherry picked one of many effects of UCA1 which might affect mRNA stability or other aspects of the regulation of most gene in erythroid cells.

Response 1:

The reviewer is correct in that there were 921 common differentially expressed genes (DEGs) after UCA1 and PTBP1 downregulation (Figure 4A), and these genes were linked to multiple biological processes and signaling pathways, including heme metabolism, G2M checkpoint, E2F targets, etc., based on GSEA-hallmark enrichment¹ (Figure 4B). These results indicated that UCA1 and PTBP1 play multiple roles in erythroid cells. Since heme

metabolism (Figure 4B and C), especially heme biosynthetic process (Figure 4E), was the most significantly enriched pathway from these common DEGs, we inferred that erythroid differentiation blockage caused by UCA1 or PTBP1 downregulation involved, at least in part, defective heme biosynthesis. In principle, the cellular processes other than heme metabolism, such as cell cycle regulation, glycolysis, etc., could contribute to UCA1 and PTBP1 regulatory functions in erythroid cells as well. The revision also included a more thorough analysis of alternative interpretations in the Results (page 16, lines 315-323; page 19, lines 366-378;) and Discussion sections (page 27, lines 535-539).

(2) Similarly, even within the heme biosynthesis pathway, the authors focused exclusively on ALAS2 which leaves the impression that this mechanism might only affect ALAS2. Author should discuss own specific they believe the mechanism to be.

Response 2:

1) The majority of genes involved in heme biosynthesis were decreased after UCA1 and PTBP1 downregulation (Figure 4D). We quantified the downregulated heme biosynthesis genes with qRT-PCR as well (Figure 4F and G) (page 19, lines 379-384).

2) Although PTBP1 RIP assay revealed that multiple transcripts associated with heme biosynthesis interacted with PTBP1 in an immortalized human umbilical cord blood CD34⁺-derived erythroid progenitor cell line HUDEP-2 cells (Figure 5A), the endogenous RNA-RNA pull-down assay using UCA1 probes demonstrated that UCA1 selectively interacts with *ALAS2* mRNA, but not other mRNAs, such as *ALAD*, *UROS*, *FECH*, that are also involved in heme biosynthesis (Figure 5B and C) (page 20, lines 386-394).

3) After *ALAS2* overexpression, we found that *ALAS2* mRNA enrichment from PTBP1

immunoprecipitates was significantly reduced in *UCA1*^{+/-} cells (Figure 6K), suggesting that *UCA1* specifically facilitates PTBP1-dependent *ALAS2* mRNA stability (page 23, lines 455-459).

(3) The binding site for PTBP1 is very small and most likely present in many mRNA. Is there any evidence that the proposed mechanism is specific for *ALAS2*.

Response 3:

1) As mentioned in response 2, what we proposed is *UCA1* specifically facilitates PTBP1-dependent *ALAS2* mRNA stability. The CUCC sequences is canonical PTBP1 binding motif, which is not specific to *ALAS2* mRNA.

2) We have demonstrated that PTBP1 interacts with *ALAS2* mRNA via its binding motif embedding at 5' *ALAS2* mRNA from 850 to 1100 nucleotides (Figure 5H-J). To further functionally assess whether the motif is relevant to the PTBP1-*ALAS2* mRNA interaction, we *in vitro* transcribed *ALAS2* mRNA with PTBP1 binding motif deleted (F4Δ(CUCC)) (Figure 5H) and incubated with cytoplasmic extracts from HUDEP-2 cells. Following immunoblotting with PTBP1 antibody, we found there was no enrichment with the mutant (F4Δ(CUCC)) fragment, although a strong signal was detected in the F4 fragment (Figure 5H and K). We conclude that PTBP1 interaction with *ALAS2* mRNA in erythroid cells requires its binding motif (page 20, lines 402-419).

Figure 2: Expression of the genes involved in heme metabolism that are believed to be regulated by *UCA1* should be shown, rather than expression of relatively irrelevant genes. Differential regulation should be demonstrated by Q-PCR

Response 4:

1) As recommended by the reviewer, we included genes involved in heme metabolism after UCA1 downregulation in Figure 4D. Differential regulation of key genes associated with heme biosynthesis were quantified by qRT-PCR (Figure 4F and G) (page 19, lines 379-384).

2) In Figure 2I, we demonstrated that although erythroid differentiation was delayed after UCA1 downregulation, key erythroid transcription factors and functional components were largely unaffected (page 17, lines 323-327)

(5)Figure 6: the data is not really convincing that GATA1 regulate UCA1. A crispr in K562 deletion of the GATA1 would be more convincing because the assay that were performed could be associate to indirect effects.

Response 5:

Because GATA1 is essential for erythroid differentiation² and our prior work has demonstrated that major manipulations of GATA1 expression in erythroblasts is deleterious to proliferation, differentiation and/or survival, we are concerned that even if a genomic deletion of GATA1 could be generated (this would create a selective disadvantage for the mutant cells), the cells were be severely damaged and not useful for conducting mechanistic analyses. Thus, as an alternative strategy, we generated a GATA site(s) mutant *UCA1* promoter and conducted luciferase reporter assays in 293T cells (with GATA1 expression) (Supplemental Figure 7C and D) and in K562 cells (with endogenous GATA1) (Figure 7C). These results revealed high luciferase activity with the wild-type *UCA1* promoter (Supplemental Figure 7D and Figure 7C). In contrast, the luciferase activity was considerably reduced with the GATA site(s)-mutant *UCA1* promoter (Supplemental Figure 7D and Figure 7C). These results indicated that GATA1 is competent to activate *UCA1* promoter-mediated transcription

(page 23, lines 470-475). In aggregate, the results from the *UCA1* promoter analyses, GATA1 loss-of-function studies (Figure 7D-I) and GATA1 ChIP analyses in primary erythroid cells (Figure 7A and B) strongly support the conclusion that GATA1 bound to the *UCA1* promoter regulates its transcription in erythroid cells (page 23, lines 460-481).

(6) Description of the RNA-seq experiments and of the bioinformatics processing is not detailed enough

As suggested by the reviewer, we expanded the description of the RNA-seq experiments and data processing in the revised manuscript of the Materials and Methods section (page 13, lines 252-267).

Reviewer #2 (Remarks to the Author):

The manuscript by Liu and colleagues detail studies on the role of *UCA1* lncRNA in erythroid cell development. The authors either primary umbilical cord-derived erythroid progenitor cell or the erythroid leukemia cell line K562 cells to first, screen for lncRNAs showing differential expression at the proerythroblast stage compared to earlier progenitor stages, and identify *UCA1* lncRNA as one of the most differentially expressed non-coding transcript during this transition. Next, the authors use a knock down approach to show a requirement for the expression of this lncRNA in erythroid differentiation. The authors next attempt to identify the interacting factors that mediate the function of this lncRNA and the rest of the manuscript is devoted to various knock down, overexpression and affinity purification experiments aiming to mechanistically elucidate the mode of action of *UCA1* in erythroid differentiation. While study of the role of lncRNAs in differentiation of blood cells

is certainly of great value and the initial observations of the authors are very interesting, unfortunately several major shortcomings strongly affect the impact of this study. Further, the manuscript is written in a highly minimalistic way, to the point that some experiments are impossible to evaluate, and it's frequently necessary to combine the main text, materials and methods and the figure legends to barely get a grasp of what the experiment entails. The data are over-interpreted and alternative hypotheses have not been considered. Many key controls are missing and common experimental procedures aiming at reducing artifactual data have been ignored. Some of these issues are detailed below, along with some additional shortcomings.

We appreciate the reviewer's constructive comments. We revised and included additional detail in the Materials and Methods section in the manuscript (page 6-14). The revision also included a more thorough analysis of alternative interpretations in the Results (page 16, lines 315-323; page 19, lines 366-378;) and Discussion sections (page 27, lines 535-539).

(1) The majority of the studies beyond the initial characterization detailed above has been carried out in K562 cells, an erythroleukemia cell line, and neither of the findings have been validated in primary cells. Thus, the majority of the manuscript is focused on the mechanism of erythroid differentiation in a cancerous cell line, which may or may not reflect the situation in primary cells. This is particularly a concern when studying the function of lncRNAs, which have an expression pattern and mechanism of function which is highly cell type and state-dependent.

Response 1:

Based on the reviewer's recommendation, we have made significant efforts to use primary erythroid cells differentiated from human cord blood CD34⁺ cells to perform mechanistic analyses. However, even when we added the cells up to 20 million differentiated primary erythroid cells per sample for RNA pull-down assay (~ 1 million CD34⁺ cells purified from 100 ml cord blood), the cell number was still too low to generate reproducible data. When including all the positive and negative controls, we need at least 100 million primary erythroid cells for a single experiment. Given the limited access of the human cord blood and the expense to procure these primary cells, this is not feasible, and accordingly very few RNA-related mechanistic studies have used primary cells such as these.

As an alternative strategy, we used an immortalized human erythroid progenitor cell line generated from the human umbilical cord blood CD34⁺ cells (HUDEP-2)³. The HUDEP cells can express erythroblast specific surface marker (CD71 and CD235a), produce sufficient amount of hemoglobin and, importantly, generate enucleated mature red blood cells after inducing erythroid differentiation. It is the best cell line available as a close mimic of primary erythroid cells. HUDEP cells have therefore been broadly used for erythroid mechanistic analyses^{4, 5, 6}.

In the revised manuscript we have conducted nearly all of the RIP and RNA pull-down assays in HUDEP-2 cells. Please see Figure 3C and D, Figure 5A-G, which were described on page 18, lines 341-351 and page 20, lines 386-397.

(2) Several key studies, including lncRNA pull down and overexpression studies, have been performed using highly artificial systems and outdated methodologies. The use of in vitro-transcribed RNAs for pull down assays, which would have been acceptable 6-7 years

ago, is currently thought of as highly error-prone, as all RNAs fold and associate with proteins in a co-transcriptional manner and thus, such *in vitro*-transcribed RNAs are frequently both misfolded and unable to properly associate with their native set of interacting proteins. Several manuscripts detailing much more reliable methodologies using pull down of endogenous lncRNAs have been published in recent years and have become the standard procedure for defining the interactome of an RNA molecule for the last several years. The authors should use these much more reliable technologies to properly identify the interactome of UCA1 in primary cells.

Response 2:

1) To validate the RNA-protein interaction by RNA pull-down followed with protein immunoblotting, we conducted the endogenous RNA pull-down in AraC-induced K562 cells, e.g. PTBP1 immunoblotting after UCA1 pull-down. While we utilized several endogenous RNA-pull down methods ^{7, 8, 9, 10, 11}, the signals were too weak to make a convincing conclusion, as shown in the following figure.

2) In order to generate rigorous and reproducible data, we conducted additional experimentation with *in vitro* transcribed RNAs. To minimize the probability of potential artifacts, all the *in vitro* RNA pull-down assays 1) included two negative controls: antisense

RNA and beads; 2) analyzed in two cell lines: in AraC-induced K562 and HUDEP-2 cells (Figure 3C and D, Figure 5).

To further reduce the probability of potential artifacts, in this study all the RIP, as well as RNA-RNA pull-down assays utilized either endogenous proteins or RNAs. (Figure 3D, Figure 5A-F). Test revised on page 18, lines 341-351 and page 20, lines 386-399.

(3) Another example of the use of an outdated methodology and lack of appropriate controls is the *in situ* hybridization studies shown in Fig. 3A. For the last several years, it has become standard procedure to use multiple probes or sets of probes, to ensure that the observed signal is not the artifact of an off-target interaction of a particular probe, along with a control, non-targeting probe. As presently performed, it's not possible to rule out such an off-target effect, or even rule out that the observed signal is not an artifact of staining.

Response 3:

We appreciate the critical comment. In the revised manuscript, we utilized a highly sensitive and specific RNA *in situ* hybridization technology with RNAscope^B Multiplex fluorescent Reagent Kit (V2)¹² to detect UCA1 expression and cellular distribution, which allows simultaneous signal amplification and background suppression to achieve single-molecular visualization. It has been utilized in >1,000 research publications since 2011^{13, 14, 15}.

To rule out the possibility of off-target effects or other artifacts, 20 sets of UCA1 probes, as well as the positive control probes from peptidylprolyl isomerase B (*PPIB*) and negative control probes from *dapB* (from *E.Coli*) were used. The bacterial gene *dapB* was used to assess background signals. The specificity of UCA1 probes was further tested in mouse

erythroleukemia cells (MELs). The results revealed that UCA1 was predominantly distributed in the cytoplasm of the day 8-differentiated erythroid cells (Figure 3A). The revised text was incorporated in Material and Methods (page 9, lines 144-160) and Results (page 17, lines 334-341) section.

(4) Have the RIP assays detailed on top of page 12 performed using whole cell extract or only cytoplasmic extracts? PTBP1 is mostly nucleoplasmic, and thus, the pull down from total cellular extracts may mostly yield information about the association of PTBP1 with the immature nuclear RNAs. Have the authors considered this possibility when performing qRT-PCR to detect UCA1, which is a spliced message? Have they used probes that flank each of the two exon-exon junctions in UCA1 to determine the share of immature versus mature UCA1 transcripts in the interaction they observe in RIP assays?

Response 4:

We utilized whole cell extracts from K562 or HUDEP-2 cells for RIP. The primers for all detected genes flanked exon-exon junctions.

(5) An example of overstating the data and ignoring alternative hypotheses is the interpretation of the PTBP1 knock down data on page 12. Knock down of proteins as critical to cellular RNA biogenesis and function as PTBP1 is expected to affect many RNAs, and thus disrupting many cellular processes, including differentiation. Until proven otherwise, such failure to differentiate phenotypes are not by any means due to a UCA1-mediated mechanism. Knock down of many other key factors in erythroid differentiation or general transcriptional and post-transcriptional regulatory mechanisms could result in failure to differentiate, with

downstream effects identical to the ones described for UCA1 and PTBP1 knock downs.

However, it does not mean that all such factors act through association with UTA1, as the authors state on page 12 and 13. In fact, this reviewer could not find a single firm, dependable data in the entire manuscript that proved that the association of UCA1 with PTBP1, assuming that it is a real interaction, has any role in erythroid differentiation beyond possibly supporting some aspect of the biogenesis of UCA1 itself which has not been shown nor investigated in this manuscript.

Response 5:

1) In this study, we demonstrated UCA1 interaction with PTBP1 using RIP and RNA pull-down assays in AraC-induced K562 and HUDEP-2 cells, respectively (Figure 3B-D and Supplemental Figure 3A and B). Text revised on page 18, lines 341-351.

2) We conducted loss-of-function analyses and identified 4,410 differentially expressed genes (DEGs) after PTBP1 knockdown (Supplemental Figure 4A). GSEA-hallmark enrichment from these DEGs were associated with heme metabolism, mTORC1 signaling, etc (Supplemental Figure 4B-C), suggesting PTBP1 has broad functions in erythroid cells. Text revised on page 19, lines 366-371.

3) UCA1 and PTBP1 common downstream targets were most significantly linked to heme biosynthesis (Figure 4A and E), suggesting that erythroid differentiation blockage caused by UCA1 or PTBP1 downregulation involved, at least in part, defective heme biosynthesis. Text revised on page 19, lines 372-378.

4) We revealed that UCA1, PTBP1 protein and *ALAS2* mRNA form a protein-RNA complex (Figure 5A-G and Supplemental Figure 5), which confers *ALAS2* mRNA stability

(Figure 6A-B). When UCA1 was depleted, even with ALAS2 overexpression (Figure 6G and H), the interaction between PTBP1 and *ALAS2* mRNA was impaired (Figure 6K), suggesting that the PTBP1-*ALAS2* mRNA interaction is UCA1-dependent. Thus, we inferred that the PTBP1 and UCA1 interactions in erythroid cells involved in heme biosynthesis, at least partially involves regulation of *ALAS2* mRNA (page 23, lines 455-459).

5) The cellular processes other than heme metabolism, such as cell cycle regulation, glycolysis, etc could contribute to UCA1 and PTBP1 regulatory functions in erythroid cells as well. We revised the manuscript accordingly and included alternative hypothesis in the discussion section (page 27, lines 535-539).

(6) In Figs. 5A and 5B, the authors attempt to show, using RIP assays and the outdated RNA pull down assays detailed in point #2 above, that both UCA1 and ALAS2 interact with PTBP1, and partly based on these results, go ahead to conclude that UCA1, ALAS2 and PTBP1 must form a ternary complex. The authors don't seem to appreciate that PTBP1 and all other hnRNPs bind a very large number of cellular transcripts especially those that are spliced (in the case of PTBP1), and thus, observing PTBP1 interaction with cellular transcripts is not by any means unusual, especially if total cellular extracts are used in such assays.

Response 6:

1) We agree with the reviewer that PTBP1 may interact with a large number of cellular transcripts. PTBP1 RIP detected multiple transcripts involved in heme biosynthesis in HUDEP-2 cells (Figure 5A).

2) When we utilized the endogenous RNA-RNA pull-down assay with UCA1 probes, we found a much higher enrichment of *ALAS2* mRNA than other genes involved into heme biosynthesis in HUDEP-2 cells (Figure 5B and C). Since we detected UCA1, PTBP1 protein and *ALAS2* mRNA interacts with each other (Figure 5A-G), we therefore infer that they form a protein-RNA complex (page 20, lines 386-401).

(7) The one experiment that could have shown some firm data supporting the claims of the authors is the one shown in Figs. 5C and D. However, there is literally NO information about how this has been done, and how the enrichment is measured. It should be mentioned that detection of such RNA-RNA interactions is by no means routine and thus, a very detailed experimental protocol should have been provided. In the absence of such information, this key data sadly remains uninterpretable.

Response 7:

We have incorporated a detailed description of the endogenous RNA-RNA pull-down methodology in Materials and Methods section (page 12, lines 216-225).

8) the middle paragraph of page 14 is very confusing. The authors state that they make 3 probes, representing different lengths of *ALAS2 mRNA*. Are these antisense probes? They state that they were used for RNA pull down analyses followed by blotting for PTBP1. I am assuming that by probes they are referring to truncated versions of *ALAS2* to see which one interacts with PTBP1 in an artificial system (in vitro-transcribed RNAs incubated in cellular extracts). While there is a strong signal in the immunoblot for PTBP1 in the lane corresponding to the full length *ALAS2*, all truncated versions, even those that the authors claim bind PTBP1, show either no signal or woefully weak signals that cannot be used to

draw any conclusions about where the binding domain(s) of PTBP1 in *ALAS2 mRNA* is(are). Since PTBP1 is an RNA-binding protein with a very short binding motif that can be easily found in most RNAs longer than 500 nucleotides, such weak binding results should be treated as expectable artifacts. Also, in such very artificial systems, it is difficult to assess whether the various RNAs incubated in cellular extract demonstrate the same level of stability, thus further complicating the interpretation of such results. No information or even mention of the relative stability of the constructs in cellular extract is given. Thus, the failure to pull down PTBP1 could be simply due to total degradation of the truncated RNAs. Do the authors have any data to disprove this?

Response 8:

1) We apologize for generating confusion. We did refer to *in vitro* transcribed truncated *ALAS2* mRNAs with different lengths (Figure 5H).

2) We revised the figures demonstrating the interaction between PTBP1 and F1 of *ALAS2* mRNA as well as PTBP1 and F4 of *ALAS2* mRNA (Figure 5I and J).

3) To interrogate whether the binding site on the F4 fragment is functional, we *in vitro* transcribed the F4 fragment with the binding motif deleted (F4 Δ (CUCC)) and performed the RNA pull-down assay with HUDEP-2 cytoplasmic extracts. This analysis did not reveal any signal from the mutant F4 Δ (CUCC) fragment, although the intense signal was shown in the F4 fragment (Figure 5H and K), indicating that the motif is essential for *ALAS2* mRNA and PTBP1 protein interaction. Text revised on page 20, lines 402-419.

(9) In Fig. 5H, the time point zero samples should not all be adjusted to one, but should be normalized to the time point zero value in the wild type sample. If the level is already

significantly decreased in the knock down samples at time point zero, the interpretation and calculation of significance of the 24 hour values will be affected.

Response 9:

We incorporated the reviewer's recommendation in the revised Figure 6C. This additional analysis confirmed the original interpretation and assessment of significance.

(10) Another example of lack of appropriate controls and following standard procedures is that in most knock down studies in the manuscript, only one shRNA is used, creating the concern that what the authors present may be an off-target artifact. This is further exacerbated by the absence of a non-targeting shRNA as a control in these studies, which is needed to ensure that the observed effects are not the result of perturbations caused by the shRNA expression and biogenesis instead of their targeting function.

Response 10:

The functional assays in this study were conducted by two shRNAs (Figure 2 and Figure 3). Then we selected the shRNA with higher knockdown efficiency to conduct the RNA-seq analysis. To minimize potential non-specific effects, all the following downstream targets were validated with two shRNAs.

The empty puromycin-resistant vector pLKO.1 was used to produce control lentiviruses and it is acceptable to use the empty vector control ^{16, 17, 18}.

(11) Can the authors explain why the reduction in the level of UCA1 RNA is so dramatic in a heterozygous knock out cell line?

Response 11:

After sequencing the upstream and downstream nucleotide sequences around the sgRNAs in this knockout cell line, in addition to one allele deletion, we found there exists an indel at the other allele (Figure 6D), which might account for the low-level transcription of UCA1. Therefore, it is a compound heterozygous knockout cell line. Text revised on page 22, lines 429-437.

(12) Similar to the point #5 above, the interpretation of the results in Fig. S5D (detailed on page 15, 2nd paragraph) is problematic. Any gene that blocks erythroid differentiation will have the same effect as what is shown in Fig. S5D. Thus, this result does not indicate a specific relationship between UCA1 and ALAS2 RNAs.

Response 12:

We agree that this could reflect indirect regulation and not indicate a specific relationship between UCA1 and *ALAS2* mRNA. However, one can conclude that *ALAS2* mRNA expression declines after UCA1 depletion. Revised text is on page 22, lines 436-437.

(13) Similarly, the results in Fig. 6B are incorrectly interpreted. The observed results are entirely expected based on the design of the experiment, since the total level of the *ALAS2* RNA in the heterozygous knock out cell line is a lot less than it is in wild type cells. Further, the in vitro-transcribed RNAs differ from the endogenous mRNA in terms of their stability which further confounds the interpretation of the observed results. Instead of using such highly artificial systems, the authors should overexpress *ALAS2* from a transgene in a living cell and repeat this experiment in multiple clones after measuring the level of overexpression while including all the needed controls.

Response 13:

We have overexpressed the *ALAS2* in the *UCA1*^{+/-} cell as suggested by the reviewer.

This analysis indicated that the interaction between PTBP1 and *ALAS2* mRNA was significantly impaired after *UCA1* depletion even with a sufficient amount of *ALAS2* mRNA (~100-fold increase) in *UCA1*^{+/-} cells. The results are present in Figure 6G, H and K (page 23, lines 455-459).

(14) Another example of the problems caused by the omission of key information in the text is Fig. 7C. From the text, it is not clear how this experiment is done at all. One has to read the figure legend and the materials and methods section and then only guess what each bar in the graph refer to. There is of course no information about how much GATA1 protein was overexpressed over the endogenous level, thus making it difficult to evaluate the significance of the results in the figure.

Response 14:

In the revised manuscript, we incorporated detailed information of how the *in vitro* luciferase reporter assay was conducted in Materials and Methods section (page 13, lines 241-251).

We conducted new experimentation to measure GATA1 overexpression levels and this data is presented in Supplemental Figure 6C. Since there is no endogenous GATA1 expression in 293T cells, we did not quantify GATA1 expression over the endogenous level.

Text revised on page 23, lines 470-475.

Minor points:

1) Page 9, 2nd paragraph, last line : the authors state that they have used a fold change threshold of 1.2 to identify differentially expressed lncRNAs. This seems an excessively low threshold which is not only unusual, but also not consistent with what is shown in Fig. 1F. Is this a typo, or another example of super-minimalism and omitting key information? Can the authors explain why they have chosen this threshold?

We have revised this presentation to include cutoffs based on $FPKM \geq 0.1$ and fold change ≥ 2 (page 15, lines 284-286).

2) Key aspects of experiments must be included in the text, rather than buried in material and methods, which is frequently skipped by readers. For example, on page 11, last paragraph, it is essential to mention that the RNA pull down assays are performed using in vitro-transcribed RNAs incubated in extracts to ensure the readers can judge the merit of the shown data.

Thank you for the constructive suggestions. In the revised manuscript, we have incorporated essential information as recommended.

3) The results in Fig. 5H suggests that *ALAS2* has a very long half-life. Is this supported by other studies?

To our knowledge, no published data has described the *ALAS2* mRNA half-life in human erythroid cells. Compared to *GATA1* mRNA, the half-life of *ALAS2* mRNA was much longer in both K562 cells and the day 14-differentiated erythroblasts from $CD34^+$ cells, as shown the following figure.

4) Page 15, two lines from the bottom. The authors need to describe what they mean by "quantitative" ChIP analysis.

We mean to quantify the GATA1 occupancy at the *UCA1* promoter in primary erythroid cells by ChIP-qPCR. To make it clear, we revised the text as “ChIP assay, followed by qPCR” (page 23, lines 465).

Reviewer #3 (Remarks to the Author):

In this study, the authors identified *UCA1* as a long non-coding RNA (lncRNA), which was upregulated specifically in the proerythroblast stage using ex vivo erythroid differentiation system of human cord blood CD34+ cells. *UCA1* is a primate-specific lncRNA. Erythroid differentiation and hemoglobin synthesis were affected by knockdown of *UCA1*, indicating that *UCA1* is required for erythroid differentiation. *UCA1* localizes mainly in the cytoplasm. The authors identified PTBP1 that interacts with *UCA1* by RNA pull-down assays with mass spectrometry analysis. Phenotype of PTBP1-knockdown cells resembles that of *UCA1*-knockdown cells, suggesting that *UCA1* functions with PTBP1. The authors found that genes associated with heme metabolism were affected by both *UCA1*-knockdown and

PTBP1-knockdown cells. In these genes, the authors focused on *Alas2*. They found that UCA1 lncRNA, PTBP1 protein and *ALAS2 mRNA* formed a complex, which stabilized *ALAS2 mRNA*. Finally, the authors found that GATA1 directly upregulates the UCA1 expression. This study provides a novel insight for primate-specific lncRNA, which is required for erythroid differentiation. The paper is well written. To strength the authors' conclusion, the following points need to be considered;

We thank the reviewer for providing insightful comments.

The authors conclude that UCA1 induces erythroid differentiation by promoting heme synthesis, especially stabilizing *ALAS2 mRNA*. To strength this conclusion, it may be appropriate for the authors to examine whether *Alas2* overexpression rescues phenotype of UCA1-knockdown cells (erythroid differentiation and hemoglobin synthesis).

Response 1:

To address this important comment, we overexpressed *ALAS2* in *UCA1^{+/-}* cells and induced the cells to undergo erythroid differentiation (Figure 6G and H). We observed more intensely red cell pellet (Supplemental Figure 6) and increased percentage of benzidine-positive cells (Figure 6I), suggesting that *ALAS2* overexpression at least partially rescued the erythroid delay resulting from UCA1 depletion. In addition, since *ALAS2* catalyzes the production of 5-amino-levulinic acid (5-ALA) in the heme biosynthetic process¹⁹, we predicted that the addition of 5-ALA can supplement the missing metabolite, thus bypassing the heme biosynthetic defect. To test this hypothesis, we infected lentiviral-mediated UCA1-targeting shRNA (#2) into cells differentiated for 4 days and

treated the cells with 5-ALA (300 μ M) at day 6. At day 14, we observed 5-ALA partially rescued the globin synthesis by increasing percentage of benzidine-positive cells in UCA1-downregulated cells (Figure 6J). Under these conditions, the erythroid cell surface markers CD71 and CD235a were not restored (data did not show). These results suggest that other UCA1-regulated downstream targets may also be important in primary erythroid cells and/or the physiological mechanism cannot be fully reconstituted in the rescue system. This is consistent with the hallmark GSEA enrichment analysis that UCA1 is involved into multiple cellular and biological processes in erythroid cells (Figure 4B). Taken together, overexpression of ALAS2 or 5-ALA addition partially rescued the phenotypes induced by UCA1 downregulation in AraC-induced K562 or primary erythroid cells (page 22, lines 429-454).

2. The level of UCA1 expression was the highest on day 8 during erythroid differentiation of CD34⁺ progenitor cell, while the induction of benzidine positive cells was further stimulated after day 8. As the authors mention in Discussion, this reviewer considers multiple mechanisms confer to the function of ALAS2 during erythroid differentiation in a stage-specific manner. Therefore, this reviewer considers that the expression profiles of UCA1, PTBP1 and ALAS2 during erythroid differentiation should be shown in a panel, in addition to the expression profile of GATA1 that regulates both UCA1 and ALAS2 expression.

Response 2:

We agree with the reviewer that multiple mechanisms govern ALAS2 function during erythroid differentiation, and it would not be surprising if such mechanisms are cell

stage-specific manner. We quantified and showed the expression of UCA1, PTBP1, ALAS2 and GATA1 in a panel (Supplemental Figure 7B) (page 23, lines 468-469).

3. In Figure 6 and S5, the author established UCA1^{+/-} cells by CRISPR/Cas9 system. UCA1 RNA levels were dramatically decreased in UCA1^{+/-} cells, although one allele of UCA1 was intact. The author need to explain how they judge one UCA1 allele was intact.

Response 3:

After sequencing the upstream and downstream nucleotide sequences around the sgRNAs in this knockout cell line, in addition to one allele deletion, we found there exists an indel at the other allele (Figure 6D), which might account for the low-level transcription of UCA1. Therefore, it is a compound heterozygous knockout cell line. Text revised on page 22, lines 429-436.

4. The authors should describe how cellular and nuclear sizes were measured in Materials and Methods. (MM)

Response 4:

We incorporated new text in Materials and Methods section with detailed description of the approach (page 7, lines 108-114).

References:

1. Subramanian A, *et al.* Gene set enrichment analysis: a knowledge-based approach for interpreting genome-wide expression profiles. *Proceedings of the National Academy of Sciences of the United States of America* **102**, 15545-15550 (2005).
2. Weiss MJ, Yu C, Orkin SH. Erythroid-cell-specific properties of transcription factor GATA-1 revealed by phenotypic rescue of a gene-targeted cell line. *Mol Cell Biol* **17**, 1642-1651 (1997).
3. Kurita R, *et al.* Establishment of immortalized human erythroid progenitor cell lines able to produce enucleated red blood cells. *PLoS One* **8**, e59890 (2013).
4. Canver MC, *et al.* BCL11A enhancer dissection by Cas9-mediated in situ saturating mutagenesis. *Nature* **527**, 192-197 (2015).
5. Norton LJ, *et al.* KLF1 directly activates expression of the novel fetal globin repressor ZBTB7A/LRF in erythroid cells. *Blood Adv* **1**, 685-692 (2017).
6. Masuda T, *et al.* Transcription factors LRF and BCL11A independently repress expression of fetal hemoglobin. *Science* **351**, 285-289 (2016).
7. Simon MD, *et al.* The genomic binding sites of a noncoding RNA. *Proceedings of the*

National Academy of Sciences of the United States of America **108**, 20497-20502

(2011).

8. Engreitz JM, *et al.* The Xist lncRNA exploits three-dimensional genome architecture to spread across the X chromosome. *Science* **341**, 1237973 (2013).
9. Chu C, *et al.* Systematic discovery of Xist RNA binding proteins. *Cell* **161**, 404-416 (2015).
10. Chu C, Quinn J, Chang HY. Chromatin isolation by RNA purification (ChIRP). *J Vis Exp*, (2012).
11. Simon MD. Capture hybridization analysis of RNA targets (CHART). *Curr Protoc Mol Biol* **Chapter 21**, Unit 21 25 (2013).
12. Wang F, *et al.* RNAscope: a novel in situ RNA analysis platform for formalin-fixed, paraffin-embedded tissues. *J Mol Diagn* **14**, 22-29 (2012).
13. Ge X, *et al.* LEAP2 Is an Endogenous Antagonist of the Ghrelin Receptor. *Cell Metab* **27**, 461-469 e466 (2018).
14. Li J, Wang Z, Chu Q, Jiang K, Li J, Tang N. The Strength of Mechanical Forces

Determines the Differentiation of Alveolar Epithelial Cells. *Dev Cell* **44**, 297-312 e295 (2018).

15. Sillman B, *et al.* Creation of a long-acting nanoformulated dolutegravir. *Nat Commun* **9**, 443 (2018).
16. Sankaran VG, *et al.* Human fetal hemoglobin expression is regulated by the developmental stage-specific repressor BCL11A. *Science* **322**, 1839-1842 (2008).
17. Hewitt KJ, *et al.* Hematopoietic Signaling Mechanism Revealed from a Stem/Progenitor Cell Cistrome. *Molecular cell* **59**, 62-74 (2015).
18. Gorrini C, *et al.* BRCA1 interacts with Nrf2 to regulate antioxidant signaling and cell survival. *J Exp Med* **210**, 1529-1544 (2013).
19. Yamamoto M, Yew NS, Federspiel M, Dodgson JB, Hayashi N, Engel JD. Isolation of recombinant cDNAs encoding chicken erythroid delta-aminolevulinate synthase. *Proceedings of the National Academy of Sciences of the United States of America* **82**, 3702-3706 (1985).

Reviewers' comments:

Reviewer #1 (Remarks to the Author):

This is an interesting manuscript describing the role of lncRNA UCA1 in the regulation of heme biosynthesis in human cultured basophilic erythroblasts.

The main claims of the paper that UCA1 regulates ALAS2 mRNA stability through interaction with PTBP1 is well supported by the data and is quite novel.

I did not have any major concerns that would put in question the validity of the main conclusions. However, there are a few technical issues and overstatement with the manuscript:

A recurring issue is the fact that the RNA-seq data is not replicated in any of the figure 1,2 and 4. I assume that the authors did not repeat these experiment because of the expense and because they validated some of the key findings by Q-RT-PCR.

I nevertheless think that unrepeated data should not be published at least not in the main text, particularly because, there has been a lot of talk about the lack of reproducibility of scientific research in general.

Another issue is that the processing of the RNA-seq data is very poorly described, to the point that it is difficult to understand some of the figure without making a lot of assumptions.

For instance in Figure 1 and S1, the RNA-seq data is not presented clearly:

Author state that the heat map represents "differentially expressed ncRNA in any pairwise comparison" and I am not sure what that mean.

I assume this might mean that this is a composite graph in which the highest differential value for each gene was chosen. For instance, if a gene is red in the day 4 column of Figure 1F (and is let say 10 times overexpressed), it means that it was 10 times higher than either at day 8, 11 or 14 . Is that correct ?)

Authors should clarify this. What happens if a gene went up and down ? what value was chosen ? Did they use any particular software to do the analysis. IF yes which parameters were used.

Also, what is the definition of a ncRNA according to the authors?

Without this information and some basic info as to how the data was processed, the figure does not mean anything.

There here is no mention of any repeat in figure 1 and S1 for the RNA-seq data. I therefore assume it was a one of kind experiment.

In my opinion unrepeated experiments should be at best in the supplementary data or simply mentioned as not shown. Since the entire manuscript is focused on one particular gene, the RNA-seq data is not critical for this figure since the Q-RT-PCR for the UCA1 gene is sufficient to justify studying this gene. The claim that there is precisely 1269 lncRNA differentially regulated during CD34 differentiation toward the erythroid lineage is strongly substantiated. Making that claim is not necessary for this manuscript.

Figure 2:

RNA-seq data is also not repeated in this figure.

There was 2 shRNA and the data seemed to have been pooled. But it is unclear what was really done to compose that figure. Does the figure show the average of both experiments, or something else ?

Figure 3A:

RNAscope data:

One picture showing three dots is not good enough to make the point that UCA1 is strictly cytoplasmic.

Need some sort of quantification.

Author should explain why they used ARA-C induced K562 cells and HUDEP and not the CD34 cells for these experiments. Was there an issue of cell number ?

Figure 4 RNA-seq is not repeated either and the cut-off used are very different from the other cut-off ? why was that . not very convincing

Conclusion from Figure 7 is also not very convincing.

All of the GSAE and pathway analysis in figures 1 to 4 without any repeat and serious statistical analysis are somewhat questionable although all three RNA-seq experiments do go in the same direction

Figure 7: To demonstrate the importance of GATA1 in regulation of UCA1 should do Crspr knock-out else the effect could be indirect since binding of GATA1 2 kb from the promoter is not a very strong evidence of regulation. Are there other genes nearby ? Not all GATA1 binding sites are important.

The claim that GATA1 regulates UCA1 is not essential for this paper and should be either, strengthened, watered down, or eliminated completely

Reviewer #2 (Remarks to the Author):

The authors have substantially revised the manuscript and addressed many of the shortcomings of the original manuscript through changes in the text and addition of a number of new data panels. While some issues remain outstanding, the revised manuscript is much stronger and scientifically more sound. One largely unaddressed issue is that of lack of suitable controls to rule out off target effects in knock down studies, which in most cases have been conducted with a single shRNA construct and lack non targeting shRNA controls. While the authors state that they have used empty vectors as controls, they need to be aware that such controls don't address off target effects or phenotypes caused by disturbances in the miRNA processing pathways caused by overexpression of the shRNA constructs. Either the authors should include additional shRNAs AND non targeting constructs in their shRNA experiments, or they should revise the text in both the results and the discussion sections to clarify that in their current form, their data cannot rule out off target effects for several of their knock down studies.

Reviewer #3 (Remarks to the Author):

The authors have conducted new experiments answering two major concerns from this reviewer. Also, the authors have improved description of the methods. This reviewer has no extra concerns.

RESPONSE TO CRITIQUES

Manuscript reference number: **NCOMMS-17-30180A-Z**.

We thank the reviewers for providing constructive comments and recommendations to further strengthen the manuscript. We have implemented both experimental and textual revisions, and the details are articulated below. The modification sections are marked in yellow.

Reviewers' comments:

Reviewer #1 (Remarks to the Author):

This is an interesting manuscript describing the role of lncRNA UCA1 in the regulation of heme biosynthesis in human cultured basophilic erythroblasts. The main claims of the paper that UCA1 regulates ALAS2 mRNA stability through interaction with PTBP1 is well supported by the data and is quite novel.

I did not have any major concerns that would put in question the validity of the main conclusions. However, there are a few technical issues and overstatement with the manuscript:

A recurring issue is the fact that the RNA-seq data is not replicated in any of the figure 1,2 and 4. I assume that the authors did not repeat these experiment because of the expense and because they validated some of the key findings by Q-RT-PCR.

I nevertheless thing that unrepeated data should not be published at least not in the main text, particularly because, there has been a lot of talk about the lack of reproducibility of scientific research in general.

Thank you for this comment. We failed to state in the prior version of the manuscript that all RNA-seq data was generated by two biological replicates. We re-plotted the heatmap to depict the gene expression profile of each replicate rather than displaying the mean expression values of the two. Please see the revised Figure 2I, Figure 4A and D, Supplemental Figure 2D and Supplemental Figure 4A. As

recommended, we transferred the ncRNA expression profiling heatmap to supplemental Figure 1A, which depicts the average expression value for the two replicates. We revised the Results on page 7, lines 86-87, and the Materials and Methods on page 30, lines 566-567.

Another issue is that the processing of the RNA-seq data is very poorly described, to the point that it is difficult to understand some of the figure without making a lot of assumptions.

We thank the reviewer for pointing this out. In the revised Material and Methods, we have incorporated a detailed description of the RNA-seq data processing, including the software and the specific parameters used for the analysis. Please see the revised text on pages 30-32, lines 575-605.

For instance in Figure 1 and S1, the RNA-seq data is not presented clearly:

Author state that the heat map represents “differentially expressed ncRNA in any pairwise comparison” and I am not sure what that mean.

I assume this might mean that this is a composite graph in which the highest differential value for each gene was chosen. For instance, if a gene is red in the day 4 column of Figure 1F (and is let say 10 times overexpressed), it means that it was 10 times higher than either at day 8, 11 or 14 . Is that correct?)

Authors should clarify this. What happens if a gene went up and down? What value was chosen? Did they use any particular software to do the analysis. IF yes, which parameters were used.

We apologize for the unintentional obscurity of our descriptions. To identify the differentially expressed ncRNAs, we collected the genes with $P_{adj} < 0.05$ for any of the pairwise comparisons [(D4 vs 8) U (D4 vs 11) U (D4 vs 14) U (D8 vs 11) U (D8 vs 14) U (D11 vs 14)]. We then identified those with an $FPKM > 0.1$ (an average of two biological replicates) at any differentiation stage (D4 U D8 U D11 U D14).

Finally, these ncRNAs were plotted based on their log-transformed FPKM values using P-heatmap (version 1.0.8) (Supplemental Figure 1A, left panel). As suggested by the reviewer, we transferred the ncRNA expression profiling heatmap to Supplemental Figure 1A, which depicts the average expression value of two replicates. We also revised text on page 7, lines 86-87 and in Supplemental Figure legend 1A on page 1, lines 2-11.

Also, what is the definition of a ncRNA according to the authors?

Without this information and some basic info as to how the data was processed, the figure does not mean anything.

The ncRNAs were defined as genes excluding all protein coding ones from GENCODE (version 27) annotation (https://www.encodegenes.org/gencode_biotypes.html). The descriptive text was revised in Supplemental Figure legend 1A on page 1, lines 4-5.

There here is no mention of any repeat in figure 1 and S1 for the RNA-seq data. I therefore assume it was a one of kind experiment.

In my opinion unrepeated experiments should be at best in the supplementary data or simply mentioned as not shown. Since the entire manuscript is focused on one particular gene, the RNA-seq data is not critical for this figure since the Q-RT-PCR for the UCA1 gene is sufficient to justify studying this gene. The claim that there is precisely 1269 lncRNA differentially regulated during CD34 differentiation toward the erythroid lineage is strongly substantiated. Making that claim is not necessary for this manuscript.

In this study, all RNA-seq data was generated through the analysis of two biological replicates. As suggested by the reviewer, we transferred the ncRNAs expression profile heatmap to Supplemental Figure 1A, which depicts the average

expression value of the two replicates. We deleted the identification of 1,269 differentially expressed ncRNAs. Please see the revised text on page 7, line 86-87.

Figure 2:

RNA-seq data is also not repeated in this figure.

There was 2 shRNA and the data seemed to have been pooled. But it is unclear what was really done to compose that figure. Does the figure show the average of both experiments, or something else?

The RNA-seq data generated from primary erythroid cells after UCA1 knockdown (control vs shUCA1 #2) involved two biological replicates, as shown in the revised Figure 2I. Due to the expense of RNA-seq, we only selected one shRNA for the transcriptomic study. However, to minimize potential non-specific effects, all of the following downstream targets were validated using 3 different shRNAs (Figure 4F). Please see the revised Figure legend 2I on page 42, lines 925-927.

Figure 3A:

RNAscope data:

One picture showing three dots is not good enough to make the point that UCA1 is strictly cytoplasmic. Need some sort of quantification.

We agree with the importance of generating quantitative data. We conducted the quantification using confocal microscopy (UltraVIEW Vox, PerkinElmer). Samples were imaged on a spinning disk confocal microscope equipped with a 100× objective and 405nm and 488nm excitation lasers. Representative cells were analyzed by generating Z stacks. Quantitative analyses and reconstruction into 3D images was

conducted using Volocity (PerkinElmer, Version 6.0). This analysis revealed that UCA1 is expressed and distributed predominantly in the cytoplasm of erythroid cells (Supplemental Figure 3A and B). The signals detected in the nucleus may derive from UCA1 RNA primary transcripts since not all probes span the exon junctions. We revised the text in Material and Methods section on page 25, lines 455-459.

Author should explain why they used ARA-C induced K562 cells and HUDEP and not the CD34 cells for these experiments. Was there an issue of cell number?

Yes, the cell number requirement was the basis for this decision. We made significant efforts to use primary erythroid cells differentiated from human cord blood CD34⁺ cells to conduct the mechanistic analyses during the prior manuscript revision. However, even when we utilized 20 million differentiated primary erythroid cells per sample for the RNA pull-down assay (~ 1 million CD34⁺ cells purified from 100 ml of cord blood), the cell number was still too low to yield reproducible data. When including all of the positive and negative controls, we needed at least 100 million primary erythroid cells for a single experiment. Given limited access to human cord blood and the considerable expense to procure these primary cells, we utilized HUDEP-2 cells instead. HUDEP-2, an immortalized human erythroid progenitor cell line generated from human umbilical cord blood CD34⁺ cells¹, is the best cell line available that mimics primary human erythroid cells. HUDEP cells can express erythroblast-specific surface markers (CD71 and CD235a), they produce abundant hemoglobin and, importantly, they generate enucleated mature red blood cells after inducing erythroid differentiation. HUDEP cells have been broadly used for erythroid mechanistic analyses^{2, 3, 4, 5}. In addition, AraC-induced human K562 cells represent another model system used to investigate biochemical mechanisms in erythroblasts.

Figure 4 RNA-seq is not repeated either and the cut-off used are very different from the other cut-off? why was that . not very convincing

To ensure consistency, we adjusted the cutoffs to $P_{adj} < 0.05$ and $FPKM > 0.1$ to identify differentially expressed genes after UCA1 or PTBP1 knockdown (Figure 2I, Figure 4 and Supplemental Figure 4). Accordingly, we also conducted the enrichment analyses with these newly identified DEGs (Figure 2J, Figure 4A-E and Supplemental Figure 4A-C). The results are entirely consistent with the previous conclusions.

All of the GSAE and pathway analysis in figures 1 to 4 without any repeat and serious statistical analysis are somewhat questionable although all three RNA-seq experiments do go in the same direction.

In this study, all RNA-seq data were generated with two biological replicates; we apologize for not making that sufficiently clear. For GO or hallmark enrichment analysis, input DEGs was generated by DESeq2 with cutoffs of $P_{adj} < 0.05$ and $FPKM > 0.1$. All P values of GO or hallmark enrichment analysis were corrected for multiple testing using the Benjamini-Hochberg method and the threshold for significant enrichment was set at $P_{adj} < 0.05$. For GSEA, the expression matrix was inputted with two biological replicates, four key statistics was performed, including enrichment score (ES), normalized enrichment score (NES), P value and false discovery rate (FDR). We use a general threshold ($P < 0.05$ and $FDR < 0.25$) for defining significant enrichment. Please see the Materials and Methods section on page 31-32, lines 583-601.

Conclusion from Figure 7 is also not very convincing.

Figure 7: To demonstrate the importance of GATA1 in regulation of UCA1 should do Crspr knock-out else the effect could be indirect since binding of GATA1 2 kb from the promoter is not a very strong evidence of regulation. Are thee other genes nearby ?
Not all GATA1 binding site are important.

The claim that GATA1 regulates UCA1 is not essential for this paper and should be either, strenghtened, watered down, or eliminated completely.

We agree with the reviewer that the results included in the manuscript constitute evidence that GATA1 regulates UCA1 expression, but additional studies are required to elucidate the mechanism. Thus, we revised the manuscript to indicate that “GATA1 occupies the UCA1 promoter and regulates UCA1 expression, suggesting a potential direct transcriptional regulatory mechanism”. The text was revised on page 16, line 283-285 and page 18, lines 324-326.

Reviewer #2 (Remarks to the Author):

The authors have substantially revised the manuscript and addressed many of the shortcomings of the original manuscript through changes in the text and addition of a number of new data panels. While some issues remain outstanding, the revised manuscript is much stronger and scientifically more sound. One largely unaddressed issue is that of lack of suitable controls to rule out off target effects in knock down studies, which in most cases have been conducted with a single shRNA construct and lack non targeting shRNA controls. While the authors state that they have used empty vectors as controls, they need to be aware that such controls don't address off target effects or phenotypes caused by disturbances in the miRNA processing pathways caused by overexpression of the shRNA constructs. Either the authors should include additional shRNAs AND non targeting constructs in their shRNA experiments, or they should revise the text in both the results and the discussion sections to clarify that in their current form, their data cannot rule out off target effects for several of their knock down studies.

As suggested by the reviewer, in the revised manuscript, we incorporated data using an additional shRNA for both UCA1 and PTBP1 (Figure 2A-H and Figure 3F-K). We also validated the UCA1 and PTBP1 key downstream targets associated with heme biosynthesis process with all 3 shRNAs, respectively, by RT-qPCR (Figure 4F and G). These new results confirm the prior conclusions.

Reviewer #3 (Remarks to the Author):

The authors have conducted new experiments answering two major concerns from this reviewer. Also, the authors have improved description of the methods. This reviewer has no extra concerns.

We really appreciate this reviewer's positive comments.

1. Kurita R, *et al.* Establishment of immortalized human erythroid progenitor cell lines able to produce enucleated red blood cells. *PLoS One* **8**, e59890 (2013).
2. Canver MC, *et al.* BCL11A enhancer dissection by Cas9-mediated in situ saturating mutagenesis. *Nature* **527**, 192-197 (2015).
3. Norton LJ, *et al.* KLF1 directly activates expression of the novel fetal globin repressor ZBTB7A/LRF in erythroid cells. *Blood Adv* **1**, 685-692 (2017).
4. Masuda T, *et al.* Transcription factors LRF and BCL11A independently repress expression of fetal hemoglobin. *Science* **351**, 285-289 (2016).
5. Vinjamur DS, Bauer DE. Growing and Genetically Manipulating Human Umbilical Cord Blood-Derived Erythroid Progenitor (HUDEP) Cell Lines. *Methods Mol Biol* **1698**, 275-284 (2018).

REVIEWERS' COMMENTS:

Reviewer #1 (Remarks to the Author):

The authors have answered most of my criticisms.

Manuscript is now quite convincing and interesting